# Wasserstein Quantum Monte Carlo:
# A Novel Approach for Solving
# the Quantum Many-Body Schrödinger Equation

**Kirill Neklyudov**
Vector Institute

**Jannes Nys**
Institute of Physics & Center for Quantum Science and Engineering
École Polytechnique Fédérale de Lausanne (EPFL)

**Luca Thiede**
Vector Institute
Univeristy of Toronto

**Juan Felipe Carrasquilla**
Vector Institute
Univeristy of Waterloo

**Qiang Liu**
UT Austin

**Max Welling**
Microsoft Research
AI4Science

**Alireza Makhzani**
Vector Institute
Univeristy of Toronto

## Abstract

Solving the quantum many-body Schrödinger equation is a fundamental and challenging problem in the fields of quantum physics, quantum chemistry, and material sciences. One of the common computational approaches to this problem is Quantum Variational Monte Carlo (QVMC), in which ground-state solutions are obtained by minimizing the energy of the system within a restricted family of parameterized wave functions. Deep learning methods partially address the limitations of traditional QVMC by representing a rich family of wave functions in terms of neural networks. However, the optimization objective in QVMC remains notoriously hard to minimize and requires second-order optimization methods such as natural gradient. In this paper, we first reformulate energy functional minimization in the space of Born distributions corresponding to particle-permutation (anti-)symmetric wave functions, rather than the space of wave functions. We then interpret QVMC as the Fisher–Rao gradient flow in this distributional space, followed by a projection step onto the variational manifold. This perspective provides us with a principled framework to derive new QMC algorithms, by endowing the distributional space with better metrics, and following the projected gradient flow induced by those metrics. More specifically, we propose "Wasserstein Quantum Monte Carlo" (WQMC), which uses the gradient flow induced by the Wasserstein metric, rather than the Fisher–Rao metric, and corresponds to *transporting* the probability mass, rather than *teleporting* it. We demonstrate empirically that the dynamics of WQMC results in faster convergence to the ground state of molecular systems.

## 1 Introduction

Access to the wave function of a quantum many-body system allows us to study strongly correlated quantum matter, starting from the fundamental building blocks. For example, the solution of the time-independent electronic Schrödinger equation provides all the chemical properties of a given atomic state, which have numerous applications in chemistry and materials design. However, obtaining the exact wave function is fundamentally challenging, with a complexity scaling exponentially with

37th Conference on Neural Information Processing Systems (NeurIPS 2023).
Correspondence to: <k.necludov@gmail.com>, <makhzani@vectorinstitute.ai>

the number of degrees of freedom. Various computational techniques have been developed in the past, including compression techniques based on Tensor Networks (White, 1992), and stochastic approaches such as Quantum Monte Carlo (QMC) (Ceperley et al., 1977). Quantum Variational Monte Carlo (QVMC) (McMillan, 1965; Ceperley et al., 1977) is a well-known subclass of the latter that can, in principle, be used to estimate the lowest-energy state (i.e. ground state) of a quantum many-body system. The method operates by parameterizing the trial wave function and minimizing the energy of the many-body system w.r.t. the model parameters.

The choice of parametric family of the trial wave function is a crucial component of the QVMC framework. Naturally, deep neural networks, being a family of universal approximators, have demonstrated promising results for quantum systems with discrete (Carleo & Troyer, 2017; Choo et al., 2020; Hibat-Allah et al., 2020), as well as continuous degrees of freedom (Pfau et al., 2020; Hermann et al., 2020; Pescia et al., 2022; Gnech et al., 2022; von Glehn et al., 2022). However, the optimization process is challenging, especially for rich parametric families of the trial wave functions. This requires the use of advanced optimization techniques that take into account the geometry of the parametric manifold. The most common technique used in QVMC is referred to as 'Stochastic Reconfiguration' (SR) (Sorella, 1998), and can be seen as the quantum version of Natural Gradient Descent (Stokes et al., 2020). While for large neural networks with up to millions of parameters, efficient and scalable implementations of SR are available (Vicentini et al., 2022), it is also possible to use approximate methods such as K-FAC (Martens & Grosse, 2015; Pfau et al., 2020). Higher order optimization techniques are considered to be essential to obtain the necessary optimization performance to accurately estimate ground states of quantum many-body Hamiltonians (see e.g. (Pescia et al., 2023; Pfau et al., 2020)). Therefore, studies of the optimization procedure are an important direction for further development of the QVMC approach.

In this paper, we consider the energy minimization dynamics as a gradient flow on the non-parametric manifold of distributions. First, as an example of the proposed methodology, we demonstrate that the imaginary-time Schrödinger equation can be described as the gradient flow under the Fisher–Rao metric on the non-parametric manifold. Then, the QVMC algorithm can be seen as a projection of this gradient flow onto a parametric manifold (see Section 3 for details). Second, the gradient flow perspective gives us an additional degree of freedom in the algorithm. Namely, we can choose the metric under which we define the gradient flow. Thus, we propose and study a different energy-minimizing objective function, which we derive as a gradient flow under the Wasserstein metric (or Wasserstein Fisher–Rao metric) (Chizat et al., 2018; Kondratyev et al., 2016).

In practice, we demonstrate that incorporating the Wasserstein metric into the optimization procedure allows for faster convergence to the ground state. Namely, we demonstrate up to 10 times faster convergence of the variance of the local energy for chemical systems. Intuitively, incorporating the Wasserstein metric regularizes the density evolution by forbidding or regularizing non-local probability mass "teleportation" (as done by Fisher–Rao metric). This might facilitate faster mixing of the MCMC running along with the density updates.

## 2 Background

### 2.1 Quantum variational Monte Carlo

Consider a quantum many-body system subject to the Hamiltonian operator, which we will assume to be of the following form,

$$H = -\frac{1}{2}\nabla_x^2 + V. \tag{1}$$

where $x$ a given many-body configuration and $V$ is the potential operator. The time-dependent Schrödinger equation determines the wave function $\psi(x, t)$ of the quantum system

$$i\frac{\mathrm{d}}{\mathrm{d}t}\psi(x, t) = H\psi(x, t) \tag{2}$$

As is often the case, we will target the stationary solutions, for which we focus on solving the time-independent Schrödinger equation

$$H\psi(x) = E\psi(x) \tag{3}$$

where $E$ is the energy of the state $\psi$. The ground state of a quantum system is obtained by solving the time-independent Schrödinger equation, by targeting the eigenstate $\psi$ of the above Hamiltonian

with the lowest eigenvalue (energy) $E$. Hereby, we must restrict the Hilbert space to wave functions that are antisymmetric under particle permutations in the case of fermionic particles, and symmetric for bosons. The latter takes into account the indistinguishability of the particles. Given the Born density $q(x) = |\psi(x)|^2$, the energy of a given quantum state can be rewritten in a functional form,

$$E[\psi] = \mathbb{E}_{q(x)}[E_{\text{loc}}(x)], \quad E_{\text{loc}}(x) := \frac{[H\psi](x)}{\psi(x)} \tag{4}$$

We will focus on the case where the Hamiltonian operator is Hermitian and time-reversal symmetric. In this case, its eigenfunctions and eigenvalues are real, and the energy can be recast into a functional of the Born probability density (see also Pfau et al. (2020), where the expressions are given in terms of $\log|\psi|$)

$$E[q] = \mathbb{E}_{q(x)}[E_{\text{loc}}(x)], \quad E_{\text{loc}}(x) = V(x) - \frac{1}{4}\nabla_x^2 \log q(x) - \frac{1}{8}\|\nabla_x \log q(x)\|^2, \tag{5}$$

under the strong condition that $q(x)$ is the Born probability density derived from an (anti-)symmetric wave function: $q(x) = \psi^2(x)$. The latter will always be tacitly assumed from hereon.

The Rayleigh–Ritz principle guarantees that the $E[q]$ is lower-bounded by the true ground-state energy of the system, i.e. $E[q] \geq E_0$, if the corresponding wave function $\psi$ is a valid state of the corresponding Hilbert space. Quantum Variational Monte Carlo (QVMC) targets ground states by parametrizing the wavefunction $\psi(x, \theta)$ and by minimizing $E[q(\theta)]$. The solution to the minimization problem $\theta_0 = \arg\min_\theta E[q(\theta)]$ is obtained by gradient-based methods using the following expression for the gradient w.r.t. parameters $\theta$

$$\nabla_\theta E[q(\theta)] = \mathbb{E}_{q(x,\theta)}\left[\left(E_{\text{loc}}(x, \theta) - \mathbb{E}_{q(x,\theta)}[E_{\text{loc}}(x, \theta)]\right)\nabla_\theta \log q(x, \theta)\right]. \tag{6}$$

In sum, the above leads to an iterative procedure in which Monte Carlo sampling is used to generate configurations from the current trial state $q(x, \theta) = \psi^2(x, \theta)$, which allows computing the corresponding energy and its parameter gradients, and to update the model accordingly. In practice, the parametric model specifies the density $q(x, \theta)$ only up to a normalization constant, i.e., it outputs $\tilde{q}(x, \theta) \propto q(x, \theta)$. However, the gradient w.r.t. $\theta$ does not depend on the normalization constant; hence, throughout the paper, we refer to the model as the normalized density $q(x, \theta)$.

## 2.2 Gradient flows under the Wasserstein Fisher–Rao metric

In the previous section, we introduced QVMC in terms of Born probability functions and formulated the problem as the minimization of a functional of probability functions constrained to a variational/parametric manifold. The latter is a more common problem often tackled in machine learning, and by forging connections between both fields, we will be able to derive an alternative to QVMC.

**Gradient Flows** In Euclidean space, we can minimize a function $f : \mathbb{R}^d \to \mathbb{R}$ by following the ODE $\frac{d}{dt}x_t = -\nabla_x f(x_t)$, which can be viewed as the continuous version of standard gradient descent. Similarly, we can minimize a functional in the space of probability distributions (or in general any Riemannian manifold), by following an ODE on this manifold. However the notion of a gradient on a manifold is more complicated, and relies on the Riemannian metric that the manifold is endowed with. Different Riemannian metrics result in different gradient flows, and consequently different optimization dynamics. For a thorough analysis of gradient flows, we refer the reader to Ambrosio et al. (2005).

**Wasserstein Fisher–Rao gradient flows** Consider the space of distributions $\mathcal{P}_2$ with finite second moment. This space can be endowed with a Wasserstein Fisher–Rao metric with the corresponding distance. In particular, the *Wasserstein Fisher–Rao* (WFR) distance (Chizat et al., 2018) is defined by extending the Benamou & Brenier (2000) dynamical optimal transport formulation by a term involving the norm of the growth rate $g_t$, and by accounting for the growth term in the modified continuity equation. Namely, the distance between probability densities $p_0$ and $p_1$ is defined as

$$\text{WFR}_\lambda(p_0, p_1)^2 := \inf_{v_t, g_t, q_t} \int_0^1 \mathbb{E}_{q_t(x)}\left[\|v_t(x)\|^2 + \lambda g_t(x)^2\right] dt, \quad \text{subj. to} \tag{7}$$

$$\frac{\partial q_t(x)}{\partial t} = -\nabla_x \cdot (q_t(x)v_t(x)) + g_t(x)q_t(x), \quad \text{and } q_0(x) = p_0(x), \ q_1(x) = p_1(x),$$

where $v_t(x)$ is the vector field defining the probability flow, $g_t(x)$ is the growth term controlling the creation and annihilation of the probability mass, and $\lambda$ is the coefficient balancing the transportation and teleportation costs. Note that by setting one of the terms to zero we get 2-Wasserstein distance ($g_t(x) \equiv 0$) and Fisher–Rao distance ($v_t(x) \equiv 0$). In Section 3, we also consider the general case of $c$-Wasserstein distance, where $c$ is a convex cost function on the tangent space.

Given a functional on this manifold, $F[q] : \mathcal{P}_2 \to \mathbb{R}$, we can define the gradient flow of the function $F$ under any Riemannian metric including the Wasserstein metric, the Fisher–Rao metric, or the Wasserstein Fisher–Rao metric. For example, the gradient flow that minimizes the functional $F[q]$ under the Wasserstein Fisher–Rao metric is given by the following PDE (which is shown with detailed derivations in Appendix A)

$$
\frac{\partial q_t}{\partial t}(x) = \underbrace{-\nabla_x \cdot \left( q_t(x) \left( -\nabla_x \frac{\delta F[q_t]}{\delta q_t}(x) \right) \right)}_{\text{the continuity equation}} - \frac{1}{\lambda} \underbrace{\left( \frac{\delta F[q_t]}{\delta q_t}(x) - \mathbb{E}_{q_t(y)}\left[ \frac{\delta F[q_t]}{\delta q_t}(y) \right] \right) q_t(x)}_{\text{growth term}},
$$
(8)

where $\delta F[q]/\delta q$ is the first-variation of of $F$ with respect to the $L_2$ metric. The physical explanation of the terms in Eq. (8) is as follows. The continuity equation defines the change of the density when the samples $x \sim q_t(x)$ follow the vector field $v_t(x) = -\nabla_x \delta F[q_t]/\delta q_t$. The second term of the PDE defines the creation and annihilation of probability mass, and is proportional to the growth field $g_t(x) = \frac{\delta F[q_t]}{\delta q_t}(x) - \mathbb{E}_{q_t(y)}\left[ \frac{\delta F[q_t]}{\delta q_t}(y) \right]$. Note that $\mathbb{E}_{q_t}[g_t] = 0$, and so while mass can be "teleported", the total mass (or probability) will remain constant. The two mechanisms can be considered independently by defining the evolution terms under the 2-Wasserstein and Fisher–Rao metrics respectively, i.e.

$$
\frac{\partial q_t}{\partial t}(x) = -\nabla_x \cdot \left( q_t(x) \left( -\nabla_x \frac{\delta F[q_t]}{\delta q_t}(x) \right) \right), \qquad \text{2-Wasserstein Gradient Flow,} \qquad (9)
$$

$$
\frac{\partial q_t}{\partial t}(x) = -\left( \frac{\delta F[q_t]}{\delta q_t}(x) - \mathbb{E}_{q_t(y)}\left[ \frac{\delta F[q_t]}{\delta q_t}(y) \right] \right) q_t(x), \qquad \text{Fisher–Rao Gradient Flow.} \quad (10)
$$

It now becomes evident that the stationary condition for all the considered PDEs is

$$
\left\| \nabla_x \frac{\delta F[q_t]}{\delta q_t}(x) \right\| = 0 \iff \frac{\delta F[q_t]}{\delta q_t}(x) \equiv \text{constant}.
$$
(11)

In Appendix A, we provide derivations illustrating that Eqs. (8) to (10) correspond to the gradient flow under the Wasserstein Fisher–Rao, Wasserstein, and Fisher–Rao metrics, respectively, and hence they all minimize $F[q]$. For detailed analysis, we refer the reader to Kondratyev et al. (2016); Liero et al. (2016).

## 3 Methodology

In Section 3.1, we first demonstrate that the imaginary-time evolution of the Schrödinger equation can be viewed as a gradient flow under the Fisher–Rao metric. Afterwards, in Section 3.2, we discuss how a density evolution can be projected to the parametric variational family and show that doing so for the Fisher–Rao gradient flow yields the QVMC algorithm. Taking this perspective, we propose the Wasserstein Quantum Monte Carlo by considering Wasserstein (and Wasserstein Fisher–Rao) gradient flows, followed by the projection onto the parametric manifold (see Section 3.3).

### 3.1 Imaginary-Time evolution as the gradient flow under the Fisher–Rao metric

The ground state of a quantum system can in theory be obtained by imaginary-time evolving any valid quantum state $\psi$ (with a non-vanishing overlap with the true ground state) to infinite times. The state is evolved according to the imaginary-time Schrödinger equation, which defines the energy-minimizing time evolution of the wavefunction $\psi_t$, and is expressed as the following PDE (which is

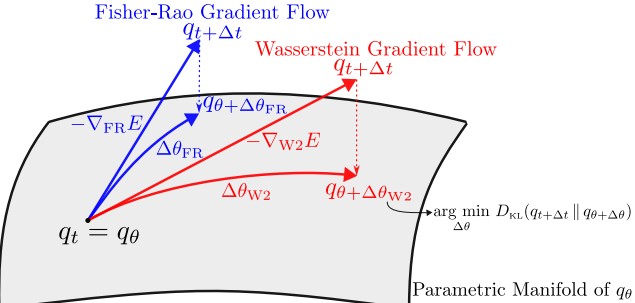

Figure 1: W(FR)QMC: A graphical illustration of the gradient flow according to the Wasserstein and Fisher–Rao metrics, and the corresponding projection onto the variational manifold $q(x, \theta)$.

the Wick-rotated version of Eq. (2), see e.g. (McArdle et al., 2019; Yuan et al., 2019)),

$$\frac{\partial \psi_t(x)}{\partial t} = -(H - E[\psi_t])\psi_t(x), \tag{12}$$

where again $q_t(x) = \psi_t^2(x)$. The last term proportional to the energy $E[\psi_t]$ comes from enforcing normalization (contrary to real-time evolution, imaginary time evolution is non-unitary).

**Theorem 3.1.** *Eq. (12) defines the gradient flow of the energy functional $\mathbb{E}[q]$ under the Fisher–Rao metric.*

*Proof Sketch.* The energy functional $E[q]$ has the following derivative

$$\frac{\delta E[q]}{\delta q}(x) = V(x) - \frac{1}{4}\nabla_x^2 \log q(x) - \frac{1}{8}\|\nabla_x \log q(x)\|^2 = E_{\text{loc}}(x). \tag{13}$$

Thus, the gradient flow under the Fisher–Rao metric is (see Eq. (10))

$$\frac{\partial q_t(x)}{\partial t} = -\left(E_{\text{loc}}(x) - \mathbb{E}_{q_t(x)}[E_{\text{loc}}(x)]\right)q_t(x), \tag{14}$$

which is equivalent (up to a multiplicative constant) to the imaginary-time Schrödinger Equation in Eq. (12) as shown in the complete proof in Appendix B. $\square$

We believe that this result can be derived following the derivations from Stokes et al. (2020), but not introducing the manifold of parametric distributions. However, considering the evolution of the density on the non-parametric manifold first helps us to derive our method and relating it to QVMC. In the following subsection, we discuss how to project this non-parametric evolution to a parametric manifold.

### 3.2 Following the gradient flow by a parametric model

By choosing a metric in the distributional space and following the energy-minimizing gradient flows, we can design various algorithms for estimating the ground state wave function. Indeed, in principle, by propagating the samples or the density according to any gradient flow (e.g., Eqs. (8) to (10)), we can eventually reach the ground state. However, these dynamics are defined on the non-parametric and infinite-dimensional manifold of distributions, which do not allow tractable computation of log densities, and thus tractable evolution. Therefore, we project these dynamics onto the parametric manifold of our variational family, and follow the *projected* gradient flows instead, which is tractable.

Suppose the current density on the parametric manifold is $q_t(x) = q(x, \theta)$ (see Figure 1). We first evolve this density using a (non-parametric) gradient flow method (e.g., Eqs. (8) to (10)) for time $\Delta t$, which will take $q_t(x)$ off the parametric manifold to $q_{t+\Delta t}(x)$. We then have to update current trial model $q(x, \theta)$ to match $q_{t+\Delta t}(x)$ enabling us to propagate the density further. In order to do so, we define the optimal update of parameters $\Delta\theta^*$ as the minimizer of the Kullback-Leibler divergence between $q_{t+\Delta t}(x)$ and the distributions on the parametric manifold, i.e.

$$\Delta\theta^* = \arg\min_{\substack{\Delta\theta \\ \|\Delta\theta\|_2=1}} D_{\text{KL}}(q_{t+\Delta t}(x)\|q(x, \theta + \Delta\theta)). \tag{15}$$

In practice, we evaluate the parameters update using the following proposition.

**Proposition 3.2.** *For $q_t(x) = q(x, \theta)$, the gradient of reverse KL-divergence can be approximated as*

$$-\nabla_\theta \mathbb{E}_{q_{t+\Delta t}}[\log q(x, \theta)] = -\int q_{t+\Delta t} \nabla_\theta \log q(x, \theta) \, \mathrm{d}x \approx -\int \left[q_t + \Delta t \frac{\partial}{\partial t} q_t\right] \nabla_\theta \log q(x, \theta) \, \mathrm{d}x$$

$$= -\int q_t \nabla_\theta \log q(x, \theta) \, \mathrm{d}x \overset{0}{\nearrow} - \Delta t \int \frac{\partial}{\partial t} q_t \nabla_\theta \log q(x, \theta) \, \mathrm{d}x \qquad (16)$$

Using this approximation, the optimal update from Eq. (15) of parameters becomes

$$\Delta \theta^* = \underset{\substack{\Delta\theta \\ \|\Delta\theta\|_2 = 1}}{\arg\min} \left\langle \Delta\theta, -\int \frac{\partial}{\partial t} q_t(x) \nabla_\theta \log q(x, \theta) \, \mathrm{d}x \right\rangle \propto \int \frac{\partial}{\partial t} q_t(x) \nabla_\theta \log q(x, \theta) \, \mathrm{d}x. \qquad (17)$$

where $\langle \cdot, \cdot \rangle$ denotes the inner product, and should not be confused with the bra-ket notation.

The following Corollary states that QVMC can be viewed as the projected gradient flow of the energy functional with respect to the Fisher–Rao metric.

**Corollary 3.3.** *Consider the Fisher–Rao gradient flow (or imaginary time evolution, which is equivalent, as shown in Theorem 3.1). Then, the parameters update (Eq. (17)) matches the gradient of the conventional QVMC loss, i.e.*

$$\Delta\theta^* \propto -\mathbb{E}_{q_t(x)}\left[\left(E_{\mathrm{loc}}(x, \theta) - \mathbb{E}_{q_t(x)}[E_{\mathrm{loc}}(x, \theta)]\right)\nabla_\theta \log q(x, \theta)\right]. \qquad (18)$$

This perspective lays the foundation for deriving our WQMC method in Section 3.3, by following the Wasserstein or WFR gradient flows rather than Fisher–Rao gradient flows.

**Natural Gradient Preconditioning**   In order to update the parametric model $q(x, \theta)$, instead of following the update in Eq. (17), we can exploit the information geometry of the statistical manifold of $q(x, \theta)$, and define the update using the Fisher information matrix $\mathcal{F}_\theta$

$$\Delta\theta^* = \underset{\substack{\Delta\theta \\ \|\Delta\theta\|_{\mathcal{F}} = 1}}{\arg\min} \left\langle \Delta\theta, -\int \frac{\partial}{\partial t} q_t(x) \nabla_\theta \log q(x, \theta) \, \mathrm{d}x \right\rangle \propto \mathcal{F}_\theta^{-1} \int \frac{\partial}{\partial t} q_t(x) \nabla_\theta \log q(x, \theta) \, \mathrm{d}x,$$

$$\mathcal{F}_\theta = \mathbb{E}_{q(x,\theta)}\left[\frac{\partial}{\partial\theta} \log q(x, \theta) \frac{\partial}{\partial\theta} \log q(x, \theta)^\top\right]. \qquad (19)$$

This update is analogous to the natural gradient update (Amari, 1998). Note that the choice of Fisher information as the metric on the statistical manifold for preconditioning the gradient and updating $\theta$ is independent of the choice of metric on the non-parametric Wasserstein manifold (e.g., Wasserstein or Fisher–Rao) for evolving $q_t(x)$. In practice, we use Kronecker-factored approximation of the natural gradient (K-FAC) (Martens & Grosse, 2015).

## 3.3   Wasserstein quantum Monte Carlo

In the previous sections, we formulated imaginary-time evolution governed by the Schrödinger equation as the energy-minimizing gradient flow under the Fisher–Rao metric. Furthermore, we demonstrated that projecting the evolved density to the parametric manifold at every iteration corresponds to the QVMC algorithm.

Naturally, we can consider another metric on the (non-parametric) space of distributions, which results in a different gradient flow and corresponds to a different algorithm. Namely, we propose to consider the gradient descent in 2-Wasserstein space as the energy-minimizing density evolution, as introduced in Section 2.2.

**Theorem 3.4.** *The energy-minimizing 2-Wasserstein gradient flow is defined by the continuity equation*

$$\frac{\partial q_t(x)}{\partial t} = -\nabla_x \cdot (q_t(x)(-\nabla_x E_{\mathrm{loc}}(x))) \qquad (20)$$

*Proof.* In Theorem 3.1, we show that $\delta E[q]/\delta q = E_{\mathrm{loc}}(x)$. Plugging this into the 2-Wasserstein gradient flow defined in Eq. (9), yields the result in Eq. (20). $\square$

$c$-**Wasserstein Metric** This result can be further generalized to the $c$-Wasserstein metric with any convex cost function $c : \mathbb{R}^d \to \mathbb{R}$ on the tangent space. The $c$-Wasserstein distance between $p_0$ and $p_1$ is defined as follows

$$W_c(p_0, p_1) := \inf_{v_t, q_t} \int_0^1 \mathbb{E}_{q_t(x)}[c(v_t(x))] \, \mathrm{d}t, \quad \text{subj. to} \tag{21}$$

$$\frac{\partial q_t(x)}{\partial t} = -\nabla_x \cdot (q_t(x) v_t(x)), \quad \text{and } q_0 = p_0, \quad q_1 = p_1. \tag{22}$$

**Proposition 3.5.** *The energy-minimizing $c$-Wasserstein gradient flow is defined by the following equation*

$$\frac{\partial q_t(x)}{\partial t} = -\nabla_x \cdot (q_t(x) \nabla c^*(-\nabla_x E_{\text{loc}}(x))), \tag{23}$$

*where $c^*(\cdot)$ is the convex conjugate function of $c(\cdot)$, and $\nabla c^*(y)$ is its gradient at $y$.*

*Proof.* See Appendix D. □

Theorem 3.4 can be viewed as a special case of Proposition 3.5 where $c(\cdot) = \frac{1}{2}\|\cdot\|^2$. Introducing a different $c$ than $L^2$ norm translates to a non-linear transformation of the gradient $-\nabla_x E_{\text{loc}}(x)$. In Appendix D, we demonstrate how to choose $c$ such that it corresponds to the coordinate-wise application of $\tanh$ to the gradient, which we use in practice.

Finally, using Proposition 3.5 in Eq. (17), we get the expression for the parameter update, i.e.

$$\Delta \theta^* \propto \int q_t(x) \nabla_\theta \left\langle \nabla c^*(-\nabla_x E_{\text{loc}}(x)), \nabla_x \log q(x, \theta) \right\rangle \mathrm{d}x. \tag{24}$$

Similar to the discussion of the previous section for QVMC, we can precondition the gradient with the Fisher Information Matrix, exploiting the geometry of the parametric manifold.

In Algorithm 1, we provide a pseudocode for the proposed algorithms. The procedure follows closely QVMC but introduces a different objective. When using gradients both from Eqs. (18) and (24), we follow the gradient flow under the Wasserstein Fisher-Rao metric with the coefficient $\lambda$. For $\lambda \to \infty$, the cost of mass teleportation becomes infinite and we use only the gradient from Eq. (24), which corresponds to the gradient flow under the $c$-Wasserstein metric (we refer to this algorithm as WQMC). For $\lambda \to 0$, the cost of mass teleportation becomes negligible compared to the transportation cost and the resulting algorithm becomes QVMC, which uses the gradient from Eq. (18). In practice, we consider the extreme cases ($\lambda \to 0, \infty$) and the mixed case $\lambda = 1$.

---

**Algorithm 1** W(FR)QMC

---

**Require:** samples $\{x^{(i)}\}_{i=1}^N \sim q_{t=0}(x)$
**Require:** potential function $V(x)$
  **while** not converged **do**
    $E_{\text{loc}}(x^{(i)}) = V(x^{(i)}) - \frac{1}{4}\nabla_x^2 \log q(x^{(i)}, \theta) - \frac{1}{8}\|\nabla_x \log q(x^{(i)}, \theta)\|^2$     (see Eq. 5)
    $\nabla_x E_{\text{loc}}(x^{(i)}) = \text{stop\_gradient}(\nabla_x E_{\text{loc}}(x^{(i)}))$
    $\Delta\theta^* = \frac{1}{N}\sum_i^N \nabla_\theta \left\langle \nabla c^*(-\nabla_x E_{\text{loc}}(x^{(i)})), \nabla_x \log q(x^{(i)}, \theta) \right\rangle$     (see Eq. 24)
    $E_{\text{loc}}(x^{(i)}) = \text{stop\_gradient}(E_{\text{loc}}(x^{(i)}))$
    $\Delta\theta^* \mathrel{+}= -\frac{1}{\lambda}\frac{1}{N}\sum_i^N \left[E_{\text{loc}}(x^{(i)}) - \frac{1}{N}\sum_j^N E_{\text{loc}}(x^{(j)})\right]\nabla_\theta \log q(x^{(i)}, \theta)$     (see Eq. 18)
    $\theta' = \text{optimizer}(\theta, \mathcal{F}_\theta^{-1}\Delta\theta^*)$     (see Eq. 19)
    update $x^{(i)}$ by sampling from $q(x, \theta')$ via MCMC
  **end while**
  **return** model $q(x, \theta^*)$, samples $\{x^{(i)}\}_{i=1}^N \sim q(x, \theta^*)$

---

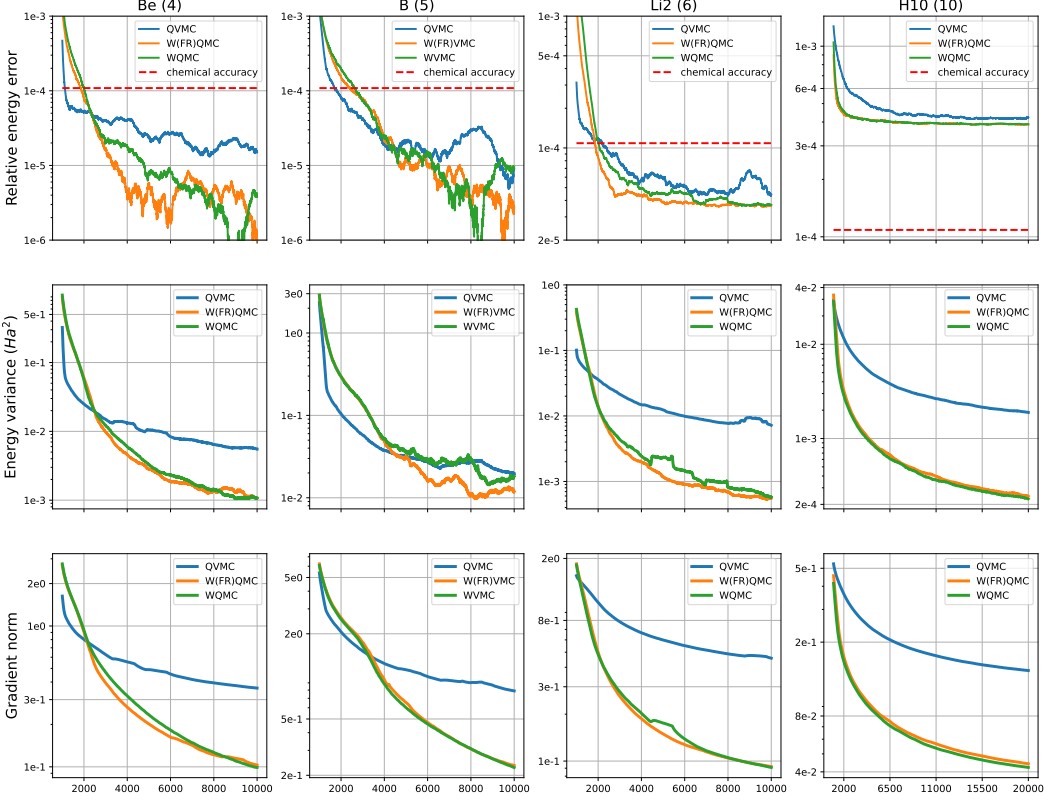

Figure 2: Optimization results for different chemical systems (every column corresponds to a given molecule). The number of electrons is given in the brackets next to systems' names. Throughout the optimization, we monitor three values: the mean value of the local energy (lower is better), the variance of the local energy, and the median value of the gradient norm of the local energy. In the first row of plots, we average (removing $5\%$ of outliers from both sides) the energy over $1000$ iterations and report the relative error to the actual ground-state energy: $(E - E_0)/E_0$. In the second row, we report standard deviation averaged over $1000$ iterations (removing $5\%$ of outliers from both sides). In the third row, we report the median gradient norm averaged over $1000$ iterations (removing $5\%$ of outliers from both sides). See the descriptions of methods in the text.

# 4 Experiments [1]

For the empirical study of the proposed method, we consider Born–Oppenheimer approximation of chemical systems. Within this approximation, the wave function of the electrons in a molecule can be studied separately from the wave function of the atomic nuclei. Namely, we consider the following Hamiltonian

$$H = -\frac{1}{2}\nabla_x^2 + \sum_{i<j}\frac{1}{\|x_i - x_j\|} - \sum_{i,I}\frac{Z_I}{\|x_i - X_I\|} + \sum_{I<J}\frac{Z_I Z_J}{\|X_I - X_J\|}, \qquad (25)$$

where $x_i$ are the coordinates of electrons, $X_I, Z_I$ are the coordinates and charges of nuclei. The first kinetic term contains derivatives with respect to the electron positions $x$. Indeed, the positions of the nuclei are given and fixed, and we target the ground state of the electronic wave function $\psi(x)$, which is an explicit function of the electron positions only. Solving the electronic Schrödinger equation is a notoriously difficult task, and is a topic of intense research in quantum chemistry and material sciences.

---

[1]Code reproducing experiments is available at github.com/necludov/wqmc. The method is also implemented in the FermiNet library: github.com/google-deepmind/ferminet.

| Method name | QVMC | WQMC | W(FR)QMC | QVMC | WQMC | W(FR)QMC |
|---|---|---|---|---|---|---|
| Molecule | | Be (4) | | | B (5) | |
| Relative energy error | 1.50e-5 | 3.79e-6 | **1.04e-6** | 9.01e-6 | 9.58e-6 | **2.69e-6** |
| Energy variance (Ha$^2$) | 5.53e-3 | 1.08e-3 | **1.07e-3** | 1.96e-2 | 1.84e-2 | **1.19e-2** |
| Molecule | | Li$_2$ (6) | | | H$_{10}$ (10) | |
| Relative energy error | 4.43e-5 | 3.71e-5 | **3.66e-5** | 4.24e-4 | 3.90e-4 | **3.88e-4** |
| Energy variance (Ha$^2$) | 7.21e-3 | 5.76e-4 | **5.59e-4** | 1.89e-3 | **2.30e-4** | 2.45e-4 |

Table 1: Energy and variances estimates for all systems after 10k iterations (20k for H$_{10}$).

Since electrons are indistinguishable fermions, we restrict the Hilbert space to states $\psi$ that are antisymmetric under electron permutations (see Section 2.1). This can be achieved by incorporating Slater determinants into the deep neural network, which parametrizes the wave function $\psi(x, \theta)$, as proposed in various recent works (Luo & Clark, 2019; Hermann et al., 2020; Pfau et al., 2020; Hermann et al., 2022). The density is then given by the Born rule $q(x, \theta) = |\psi(x, \theta)|^2$. For all our experiments, we follow (von Glehn et al., 2022) and use the "psiformer" architecture together with preconditioning the gradients via K-FAC (Martens & Grosse, 2015).

In our method, we apply several tricks which stabilize the optimization and improve convergence speed. Firstly, we have observed that applying a $\tanh$ non-linearity coordinate-wise to the gradient $\nabla_x E_{\text{loc}}(x)$ significantly improves convergence speed. This corresponds to a different cost function in the Wasserstein metric, as we discuss in Proposition 3.5 and Appendix D. Also, we remove samples from the batch whose norm $\|\nabla_x \log q(x, \theta)\|$ significantly exceeds the median value. Namely, we estimate the deviation from the norm as $\mathbb{E}_{q(x,\theta)}[|\|\nabla_x \log q(x, \theta)\| - \text{median}(\|\nabla_x \log q(x, \theta)\|)|]$ and remove samples whose norm exceeds five deviations from the median. When including the gradient from Eq. (18), we clip the local energy values as proposed in (von Glehn et al., 2022), i.e. by estimating the median value and clipping to five deviations from the median, where the deviation is estimated in the same way as for the norm of the gradient.

We consider different chemical systems and compare against QVMC as a baseline. We run our novel method with the same architecture and hyperparameters as the baseline QVMC-based approach in (von Glehn et al., 2022). For the chemical systems, we consider Be, and B atoms, the Li$_2$ molecule and the hydrogen chain H$_{10}$ from (Hermann et al., 2020). The exact values of energies for Be, B, Li$_2$ are taken from (Pfau et al., 2020), the exact value of the energy for H$_{10}$ is from (Hermann et al., 2020). All the hyperparameters and architectural details are provided in the supplementary material.

In Figure 2, we demonstrate the convergence plots for the baseline (QVMC) and the proposed methods (WQMC and W(FR)QM, see Algorithm 1). For all the considered systems, both WQMC and W(FR)QMC yield more precise estimations of the ground state energy (the first row of Figure 2). To assess convergence, we also monitor the variance of the local energy and the gradient norm of the local energy. As we discuss in Eq. (11), both metrics must vanish at the ground state. More fundamentally, the variance of the local energy can be shown to vanish for eigenstates of the Hamiltonian (Wu et al., 2023), referred to as the zero-variance property. First, we point out that obtaining the ground states of the considered molecules with QVMC is challenging, and even with powerful deep-learning architecture, discrepancies remain with the ground state. Since we use existing state-of-the-art architectures as a backbone, our results are also limited by the limitations of the latter (Gao & Günnemann, 2023). Developing novel architectures is out of the scope of this work.

However, in Figure 2, we clearly observe that both WQMC and W(FR)QMC yield significantly faster convergence of the aforementioned metrics compared to QVMC. In particular, for the larger molecules Li$_2$ and H$_{10}$, we observe that we consistently obtain lower energies within 10k steps (20k for H$_{10}$) with a more stable convergence. For the smaller molecules we observe that QVMC obtains lower energies in the first few iterations, but its convergence slows down significantly, after which our approach steadily yields improved energies below QVMC. Overall, our experiments demonstrates that taking into account the Wasserstein metric allows for faster convergence to accurate approximations of the ground state. See the final metrics in Table 1.

## 5 Limitations

Since our method requires an extra gradient evaluation compared to QVMC, we include the runtime of all the algorithms in Figure 3, Appendix E. Namely, for a proper comparison, instead of reporting the metrics per iteration, we report them per wall time in seconds. Note that all the claims of the experimental section still hold in terms of wall time. All the models were benchmarked on four A40 GPUs. Third-order derivatives can be efficiently computed using modern deep learning frameworks such as JAX (Bradbury et al., 2018), which we used to implement our method.

Potentially, one can alleviate the extra cost of the iteration by coming up with more efficient Monte Carlo schemes or other updates of the samples. Indeed, in our experiments, we observed that the proposed method requires fewer MCMC steps (not included in the paper). Moreover, one can use the evaluated gradient of the local energy as the proposal vector field for updating the samples. This would allow to decrease the number of MCMC steps at the low cost of additional hyperparameter tuning.

## 6 Discussion and conclusion

**Conclusion**    In the current paper, we propose a novel approach to solving the quantum many-body Schrödinger equation, by incorporating the Wasserstein metric on the space of Born distributions. Compared to the Fisher–Rao metric, which allows for probability mass "teleportation", the Wasserstein metric constrains the evolution of the density to local changes under the locally-optimal transportation plan, i.e., following fluid dynamics. This property is favorable when the evolution of the parametric model is accompanied by the evolution of samples (performed by an MCMC algorithm). Indeed, by forbidding or regularizing non-local mass "teleportation" in the density change, one prevents the appearance of distant modes in the density model, which would require longer MCMC runs for proper mixing of the samples.

In practice, we demonstrate that following the gradient flow under the Wasserstein (or Wasserstein Fisher–Rao) metric results in better convergence to the ground state wave function. This is expected to be due to our proposed loss, which takes into account the gradient of the local energy and achieves its minimum when the norm of the gradient vanishes, therefore explicitly minimizing the norm of the local energy gradient.

We believe that our new theoretical framework for solving the time-independent Schrödinger equation for time-reversal symmetric Hamiltonians based on optimal transport will open new avenues to develop improved numerical methods for quantum chemistry and physics.

**Connection to Energy-Based and Score-Based Generative Models**    The developed ideas of this paper, i.e., projecting gradient flows under different metrics onto a parametric family, can be extended to generative modeling by swapping the energy functional with the KL-divergence. More precisely, as we show in Appendix C, using the KL-divergence as our objective functional, the Fisher–Rao gradient flow yields energy-based training scheme, while the 2-Wasserstein gradient flow corresponds to the score-matching, which is used for training diffusion generative models.

## 7 Acknowledgement

The authors thank Rob Brekelmans and anonymous reviewers for helpful discussions and feedback. J.N. was supported by Microsoft Research. J.C. acknowledges support from the Natural Sciences and Engineering Research Council (NSERC), the Shared Hierarchical Academic Research Computing Network (SHARCNET), Compute Canada, and the Canadian Institute for Advanced Research (CIFAR) AI Chairs program. A.M. acknowledges support from the Canada CIFAR AI Chairs program. Resources used in preparing this research were provided, in part, by the Province of Ontario, the Government of Canada through CIFAR, and companies sponsoring the Vector Institute www.vectorinstitute.ai/#partners.

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

# A Gradient flows under Wasserstein Fisher–Rao metric

We, first, remind the concept of the functional derivative. The change of the functional $F[q] : \mathcal{P}_2 \to \mathbb{R}$ along the direction $h$ can be expressed as

$$F[q + h] = F[q] + \mathrm{d}F[h] + o(\|h\|), \quad \mathrm{d}F[h] = \int \mathrm{d}x \, h(x) \underbrace{\frac{\delta F[q]}{\delta q}(x)}_{\text{derivative}}. \tag{26}$$

Consider a change of the density in time, the change of the functional can be defined through the differential as

$$F\left[q_t + \Delta t \frac{\partial q_t}{\partial t}\right] = F[q_t] + \Delta t \cdot \mathrm{d}F\left[\frac{\partial q_t}{\partial t}\right] + o(\|h\|), \quad \mathrm{d}F\left[\frac{\partial q_t}{\partial t}\right] = \int \mathrm{d}x \, \frac{\partial q_t(x)}{\partial t} \frac{\delta F[q_t]}{\delta q_t}(x). \tag{27}$$

In particular, we have

$$\frac{\mathrm{d}}{\mathrm{d}t} F[q_t] = \mathrm{d}F\left[\frac{\partial q_t}{\partial t}\right] = \int \mathrm{d}x \, \frac{\partial q_t(x)}{\partial t} \frac{\delta F[q_t]}{\delta q_t}(x). \tag{28}$$

## A.1 Minimizing movement scheme for 2-Wasserstein distance and Kullback-Leibler divergence

**Gradient flow under $W_2$**   Consider the following minimizing movement scheme (MMS)

$$\inf_{q'} F[q'] - F[q] + \frac{1}{\Delta t} \frac{1}{2} W_2^2(q, q'), \tag{29}$$

where the change of the density is restricted to the continuity equation, i.e.,

$$\frac{\partial q_t}{\partial t} = -\nabla_x \cdot (q_t(x) v_t(x)), \quad \text{and} \quad q'(x) = q_t(x) - \Delta t \nabla_x \cdot (q_t(x) v_t(x)) + o(\Delta t). \tag{30}$$

Using the static formulation of $W_2$ distance, we have

$$W_2^2(q, q') = \int \mathrm{d}x \, q(x) \|x - T^*(x)\|^2 = \Delta t^2 \int \mathrm{d}x \, q(x) \|v^*(x)\|^2, \tag{31}$$

where $T^*(x)$ is the optimal transportation plan, and $v^*(x)$ is the corresponding optimal gradient field. Thus, we can rewrite the MMS problem as

$$\inf_v F[q] - \Delta t \int \mathrm{d}x \, \nabla_x \cdot (q(x) v(x)) \frac{\delta F[q_t]}{\delta q_t}(x) - F[q] + \frac{\Delta t}{2} \int \mathrm{d}x \, q(x) \|v(x)\|^2, \tag{32}$$

$$\inf_v \int \mathrm{d}x \, q(x) \left\langle v(x), \nabla_x \frac{\delta F[q_t]}{\delta q_t}(x) \right\rangle + \frac{1}{2} \int \mathrm{d}x \, q(x) \|v(x)\|^2, \tag{33}$$

$$\inf_v \int \mathrm{d}x \, q(x) \left\| v(x) + \nabla_x \frac{\delta F[q_t]}{\delta q_t}(x) \right\|^2. \tag{34}$$

From the last optimization problem, we have

$$v(x) = -\nabla_x \frac{\delta F[q_t]}{\delta q_t}(x). \tag{35}$$

**Gradient flow under KL**   Consider the following minimizing movement scheme (MMS)

$$\inf_{q'} F[q'] - F[q] + \frac{1}{\Delta t} D_{\mathrm{KL}}(q' \| q), \tag{36}$$

where the change of the density is restricted to the following weighting scheme

$$\frac{\partial q_t}{\partial t} = g_t(x) q_t(x), \quad \text{hence}, \tag{37}$$

$$q'(x) = q_t(x) + \Delta t \, q_t(x) g_t(x) + o(\Delta t), \quad \text{and} \tag{38}$$

$$\log q'(x) = \log q_t(x) + \Delta t \, g_t(x) - \frac{\Delta t^2}{2} g_t^2(x) + o(\Delta t^2). \tag{39}$$

The KL-divergence is then

$$D_{\mathrm{KL}}(q'\|q_t) = \int \mathrm{d}x \, q'(x)\left(\Delta t \, g_t(x) - \frac{\Delta t^2}{2}g_t^2(x)\right) + o(\Delta t^2) \tag{40}$$

$$= \Delta t \int \mathrm{d}x \, q_t(x)g_t(x) - \frac{\Delta t^2}{2}\int \mathrm{d}x \, q_t(x)g_t^2(x) \tag{41}$$

$$+ \Delta t^2 \int \mathrm{d}x \, q_t(x)g_t^2(x) + o(\Delta t^2) \tag{42}$$

$$= \frac{\Delta t^2}{2}\int \mathrm{d}x \, q_t(x)g_t^2(x) + o(\Delta t^2)\,. \tag{43}$$

In the last equation we are using

$$\int q_t g_t \, \mathrm{d}x = 0\,. \tag{44}$$

Thus, we can rewrite the MMS problem as

$$\inf_g F[q_t] + \Delta t \int \mathrm{d}x \, g_t(x)q_t(x)\frac{\delta F[q_t]}{\delta q_t}(x) - F[q_t] + \frac{\Delta t}{2}\int \mathrm{d}x \, q_t(x)g_t(x)^2, \tag{45}$$

$$\inf_g \int \mathrm{d}x \, q_t(x)g_t(x)\left(\frac{\delta F[q_t]}{\delta q_t}(x) - \mathbb{E}_{q_t(y)}\left[\frac{\delta F[q_t]}{\delta q_t}(y)\right]\right) + \frac{1}{2}\int \mathrm{d}x \, q_t(x)g_t(x)^2, \tag{46}$$

$$\inf_g \int \mathrm{d}x \, q_t(x)\left[g_t(x) + \left(\frac{\delta F[q_t]}{\delta q_t}(x) - \mathbb{E}_{q_t(y)}\left[\frac{\delta F[q_t]}{\delta q_t}(y)\right]\right)\right]^2. \tag{47}$$

From the last optimization problem, we have

$$g_t(x) = -\left(\frac{\delta F[q_t]}{\delta q_t}(x) - \mathbb{E}_{q(y)}\left[\frac{\delta F[q_t]}{\delta q_t}(y)\right]\right). \tag{48}$$

Note, however, that $D_{\mathrm{KL}}(\cdot\|\cdot)$ is not the same as the Fisher–Rao metric. The derivations here demonstrate that the Fisher–Rao gradient flow can be derived as the MMS scheme with the KL-divergence.

### A.2 Minimizing movement scheme for the Wasserstein Fisher–Rao metric

Consider the Wasserstein Fisher–Rao distance

$$\mathrm{WFR}_\lambda(p_0, p_1)^2 := \inf_{v_t, g_t, q_t}\int_0^1 \mathbb{E}_{q_t(x)}\left[\|v_t(x)\|^2 + \lambda g_t(x)^2\right]\mathrm{d}t, \quad \text{subj. to} \tag{49}$$

$$\frac{\partial q_t(x)}{\partial t} = -\nabla_x \cdot (q_t(x)v_t(x)) + g_t(x)q_t(x), \quad q_0(x) = p_0(x), \quad q_1(x) = p_1(x)\,. \tag{50}$$

The minimizing movement scheme (MMS) for this distance is

$$\inf_{q'} F[q'] - F[q] + \frac{1}{\Delta t}\frac{1}{2}\mathrm{WFR}_\lambda(q, q')^2, \tag{51}$$

where the change of the density is given by the continuity equation with the growth term

$$\frac{\partial q_t}{\partial t} = -\nabla_x \cdot (q_t(x)v_t(x)) + g_t(x)q_t(x). \tag{52}$$

For close enough $q$ and $q'$, $\mathrm{WFR}_\lambda(q, q')^2$ can be estimated via the metric derivative $|\mu_t'|^2$ that is defined as

$$|\mu_t'|^2 = \left(\lim_{\Delta t \to 0}\frac{\mathrm{WFR}_\lambda(q_t, q_{t+\Delta t})}{\Delta t}\right)^2, \tag{53}$$

hence,

$$\mathrm{WFR}_\lambda(q, q')^2 = \Delta t^2|\mu_t'|^2 = \Delta t^2 \int \mathrm{d}x \, q(x)\left[\|v^*(x)\|^2 + \lambda g^*(x)^2\right]. \tag{54}$$

Thus, the MMS problem can be written as

$$\inf_{g_t, v_t} F[q_t] - \Delta t \int dx \, \nabla_x \cdot (q_t(x)v_t(x)) \frac{\delta F[q_t]}{\delta q_t}(x) + \int dx \, q_t(x)g_t(x)\frac{\delta F[q_t]}{\delta q_t}(x) - F[q_t]$$
$$+ \frac{\Delta t}{2} \int dx \, q_t(x)\left[ \|v_t(x)\|^2 + \lambda g_t(x)^2 \right], \tag{55}$$

$$\inf_{g_t, v_t} \int dx \, q_t(x)\left\langle v_t(x), \nabla_x \frac{\delta F[q_t]}{\delta q_t}(x) \right\rangle + \int dx \, q_t(x)g_t(x)\left( \frac{\delta F[q_t]}{\delta q_t}(x) - \mathbb{E}_{q_t(y)}\left[ \frac{\delta F[q_t]}{\delta q_t}(y) \right] \right)$$
$$+ \frac{1}{2} \int dx \, q_t(x)\left[ \|v_t(x)\|^2 + \lambda g_t(x)^2 \right], \tag{56}$$

$$\inf_{g_t, v_t} \int dx \, q_t(x)\left\| v_t(x) + \nabla_x \frac{\delta F[q_t]}{\delta q_t}(x) \right\|^2$$
$$+ \lambda \int dx \, q_t(x)\left( g_t(x) + \frac{1}{\lambda}\left( \frac{\delta F[q_t]}{\delta q_t}(x) - \mathbb{E}_{q_t(y)}\left[ \frac{\delta F[q_t]}{\delta q_t}(y) \right] \right) \right)^2. \tag{57}$$

From the last optimization problem, we have

$$v_t(x) = -\nabla_x \frac{\delta F[q_t]}{\delta q_t}(x), \quad g_t(x) = -\frac{1}{\lambda}\left( \frac{\delta F[q_t]}{\delta q_t}(x) - \mathbb{E}_{q_t(y)}\left[ \frac{\delta F[q_t]}{\delta q_t}(y) \right] \right). \tag{58}$$

Note that different values of $\lambda$ result in different gradient flows. For instance, considering the limit $\lambda \to \infty$ we have $g(x) \to 0$ and Eq. (52) just becomes $W_2$ gradient flow, which is natural since we have an infinite penalty for the mass teleportation in our metric. Setting $\lambda \to 0$ requires some additional consideration, since then the growth term explodes, and all the mass will be teleported without any cost. Indeed, for $\lambda \to 0$, our metric does not penalize for the mass teleportation at all, but our change of density (Eq. (52)) is still able to teleport mass, hence, it will be doing so "for free".

### A.3 PDEs demonstrating the convergence

Consider the change of the density $q_t$ under the continuity equation with the vector field $v_t(x)$ and the growth term $g_t(x)$

$$\frac{\partial q_t}{\partial t}(x) = -\nabla_x \cdot (q_t(x)v_t(x)) + g_t(x)q_t(x). \tag{59}$$

Thus, the change of the functional $F[q]$ is

$$\frac{d}{dt}F[q_t] = -\int dx \, \nabla_x \cdot (q_t(x)v_t(x))\frac{\delta F[q_t]}{\delta q_t}(x) + \int dx \, q_t(x)g_t(x)\frac{\delta F[q_t]}{\delta q_t}(x) \tag{60}$$

$$= \int dx \, q_t(x)\left\langle v_t(x), \nabla_x \frac{\delta F[q_t]}{\delta q_t}(x) \right\rangle + \int dx \, q_t(x)g_t(x)\frac{\delta F[q_t]}{\delta q_t}(x). \tag{61}$$

From this equation, we can clearly see that $v_t(x)$ and $g_t(x)$ derived in the previous section minimize $F[q]$. Indeed, taking

$$v_t(x) = -\nabla_x \frac{\delta F[q_t]}{\delta q_t}(x), \quad g_t(x) = -\frac{1}{\lambda}\left( \frac{\delta F[q_t]}{\delta q_t}(x) - \mathbb{E}_{q_t(y)}\left[ \frac{\delta F[q_t]}{\delta q_t}(y) \right] \right), \tag{62}$$

we get

$$\frac{d}{dt}F[q_t] = -\int dx \, q_t(x)\left\| \nabla_x \frac{\delta F[q_t]}{\delta q_t}(x) \right\|^2 - \frac{1}{\lambda}\int dx \, q_t(x)\left( \frac{\delta F[q_t]}{\delta q_t}(x) - \mathbb{E}_{q_t(y)}\frac{\delta F[q_t]}{\delta q_t}(y) \right)^2$$

$$\tag{63}$$

$$- \frac{1}{\lambda}\underbrace{\int dx \, q_t(x)\left( \frac{\delta F[q_t]}{\delta q_t}(x) - \mathbb{E}_{q_t(y)}\left[ \frac{\delta F[q_t]}{\delta q_t}(y) \right] \right)}_{=0} \mathbb{E}_{q_t(z)}\left[ \frac{\delta F[q_t]}{\delta q_t}(z) \right] \le 0. \tag{64}$$

Note that the considered growth term preserves the normalization of the density, i.e.,

$$\int \mathrm{d}x \, \frac{\partial q_t}{\partial t}(x) = \int \mathrm{d}x \, g_t(x) q_t(x) = - \int \mathrm{d}x \, q_t(x) \left( \frac{\delta F[q_t]}{\delta q_t}(x) - \mathbb{E}_{q_t(y)}\left[ \frac{\delta F[q_t]}{\delta q_t}(y) \right] \right) = 0. \tag{65}$$

Thus, our functional $F[q]$ decreases when the density $q_t$ evolves according to the PDE

$$\frac{\partial q_t}{\partial t}(x) = \underbrace{-\nabla_x \cdot \left( q_t(x) \left( -\nabla_x \frac{\delta F[q_t]}{\delta q_t}(x) \right) \right)}_{\text{the continuity equation}} - \underbrace{\frac{1}{\lambda} \left( \frac{\delta F[q_t]}{\delta q_t}(x) - \mathbb{E}_{q_t(y)}\left[ \frac{\delta F[q_t]}{\delta q_t}(y) \right] \right) q_t(x)}_{\text{growth term}}, \tag{66}$$

and reaches its stationary point when $\left\| \nabla_x \frac{\delta F[q_t]}{\delta q_t}(x) \right\|^2 = 0$, i.e., $\frac{\delta F[q_t]}{\delta q_t}(x) \equiv \text{constant}$.

Note, that in the same way, we can consider the continuity equation and the growth term separately, which defines the gradient flows under 2-Wasserstein and Fisher–Rao metrics respectively. The corresponding PDEs are

$$\frac{\partial q_t}{\partial t}(x) = -\nabla_x \cdot \left( q_t(x) \left( -\nabla_x \frac{\delta F[q_t]}{\delta q_t}(x) \right) \right), \tag{67}$$

$$\frac{\partial q_t}{\partial t}(x) = -\left( \frac{\delta F[q_t]}{\delta q_t}(x) - \mathbb{E}_{q_t(y)}\left[ \frac{\delta F[q_t]}{\delta q_t}(y) \right] \right) q_t(x). \tag{68}$$

## B  Imaginary-time Schrödinger equation as the gradient flow under Fisher–Rao Metric

**Theorem.** *Eq.* (12) *defines the gradient flow of the energy functional* $\mathbb{E}[q]$ *under the Fisher–Rao metric.*

*Proof.* First, we derive the functional derivative of the energy functional $E[q]$. We denote the differential of the functional $F(q)$ along the direction $h$ as

$$\mathrm{d}F(q)[h] = \int \mathrm{d}x \, h \cdot \frac{\delta F[q]}{\delta q}, \tag{69}$$

where $\delta F[q]/\delta q$ is the functional derivative.

Consider the energy functional

$$E[q] = \int \mathrm{d}x \, q \left[ V - \frac{1}{4} \nabla_x^2 \log q - \frac{1}{8} \| \nabla_x \log q \|^2 \right]. \tag{70}$$

The functional derivative of this functional is as follows

$$\mathrm{d}E(q)[h] = \frac{\partial E(q + \varepsilon \cdot h)}{\partial \varepsilon} \bigg|_{\varepsilon = 0}$$
$$= \int \mathrm{d}x \, h \left[ V - \frac{1}{4} \nabla_x^2 \log q - \frac{1}{8} \| \nabla_x \log q \|^2 \right] - \int \mathrm{d}x \, q \left[ \frac{1}{4} \nabla_x^2 \frac{h}{q} + \frac{1}{4} \langle \nabla_x \log q, \nabla_x \frac{h}{q} \rangle \right].$$

For the last term, we do integration by parts and get

$$\int \mathrm{d}x \, q \left[ \frac{1}{4} \nabla_x^2 \frac{h}{q} + \frac{1}{4} \langle \nabla_x \log q, \nabla_x \frac{h}{q} \rangle \right] = -\frac{1}{4} \int \mathrm{d}x \, \left\langle \nabla_x q, \nabla_x \frac{h}{q} \right\rangle + \frac{1}{4} \int \mathrm{d}x \, \left\langle \nabla_x q, \nabla_x \frac{h}{q} \right\rangle = 0. \tag{71}$$

Thus, we have

$$\mathrm{d}E(q)[h] = \int \mathrm{d}x \, h \underbrace{\left[ V - \frac{1}{4} \nabla_x^2 \log q - \frac{1}{8} \| \nabla_x \log q \|^2 \right]}_{\delta E[q]/\delta q}, \tag{72}$$

and we see that the derivative coincides with the local energy, i.e.,

$$\frac{\delta E[q]}{\delta q}(x) = E_{\text{loc}}(x) = V(x) - \frac{1}{4}\nabla_x^2 \log q(x) - \frac{1}{8}\|\nabla_x \log q(x)\|^2. \tag{73}$$

Using the results from Section 2.2, the energy-minimizing gradient flow under Fisher–Rao metric is

$$\frac{\partial q_t(x)}{\partial t} = -\left[E_{\text{loc}}(x) - E[q_t]\right]q_t(x). \tag{74}$$

Second, we derive the PDE for the time-evolution of the density $q_t$ under the imaginary-time Schrödinger equation.

$$\frac{\partial \psi_t}{\partial t} = \frac{1}{2}\nabla_x^2 \psi_t - (V - E[q_t])\psi_t \tag{75}$$

$$2\psi_t \frac{\partial \psi_t}{\partial t} = \psi_t \nabla_x^2 \psi_t - 2(V - E[q_t])\psi_t^2 \tag{76}$$

$$\frac{\partial q_t}{\partial t} = \psi_t \nabla_x^2 \psi_t - 2(V - E[q_t])q_t \tag{77}$$

$$\tag{78}$$

Using the identity

$$\psi\nabla_x^2\psi = \psi\nabla_x \cdot (\psi\nabla_x \log|\psi|) = \langle\psi\nabla_x\psi, \nabla_x \log|\psi|\rangle + \psi^2\nabla_x^2 \log|\psi| \tag{79}$$

$$= \frac{1}{4}\langle\nabla_x q, \nabla_x \log q\rangle + \frac{1}{2}q\nabla_x^2 \log q = \frac{1}{4}q\|\nabla_x \log q\|^2 + \frac{1}{2}q\nabla_x^2 \log q, \tag{80}$$

we have

$$\frac{\partial q_t}{\partial t} = -2\left[V - \frac{1}{4}\nabla_x^2 \log q - \frac{1}{8}\|\nabla_x \log q_t\|^2 - E[q_t]\right]q_t \tag{81}$$

$$\frac{\partial q_t(x)}{\partial t} = -2\left[E_{\text{loc}}(x) - E[q_t]\right]q_t(x), \tag{82}$$

which is equivalent to Eq. (74). □

## C   Analogies to generative modeling and variational inference literature

To draw a connection to generative models and variational inference literature, we consider the reverse and forward KL-divergences as an objective to minimize (instead of the energy) and derive projected gradient flows for them.

### C.1   Generative modeling: reverse KL divergence functional

Consider the reverse KL divergence objective, and its first variation

$$\mathcal{F}[q_t] = D_{\text{KL}}(p \,\|\, q_t) \tag{83}$$

$$\implies \frac{\delta\mathcal{F}}{\delta q_t} = -\frac{p}{q_t}, \tag{84}$$

where $p(x)$ is the data distribution given empirically.

**Fisher–Rao Gradient Flow**   Using equations Eq. (10), the PDE that defines the Fisher–Rao gradient flow is

$$\frac{\partial q_t}{\partial t}(x) = \left(\frac{p(x)}{q_t(x)} - \mathbb{E}_{q_t(y)}\left[\frac{p(y)}{q_t(y)}\right]\right)q_t(x) = p(x) - q_t(x). \tag{85}$$

Thus we have

$$q_t = \exp(-t)q_0 + (1 - \exp(-t))p, \tag{86}$$

where $q_0$ is the initial distribution. This results in a distributional path constructed by the linear average (mixture) of the end-point distributions. Note that changing the mixture parameter corresponds to *teleporting* the particles.

**Projected Fisher–Rao Gradient Flow**   Having the functional minimizing PDEs, we can find the update corresponding to the projected Fisher–Rao gradient flow using Proposition 3.2 and Eq. (17),

$$\Delta\theta^* \propto -\int q_t \left(\frac{\delta\mathcal{F}}{\delta q_t}\right) \nabla_\theta \log q(x,\theta)\, \mathrm{d}x \tag{87}$$

$$= -\int q_t \left(-\frac{p}{q_t}\right) \nabla_\theta \log q(x,\theta)\, \mathrm{d}x \tag{88}$$

$$= \nabla_\theta \mathbb{E}_{p(x)} \Big[\log q(x,\theta)\Big]. \tag{89}$$

This corresponds to the standard maximum likelihood objective that is used for training of energy-based models (Xie et al., 2016; Du & Mordatch, 2019).

**Wasserstein Gradient Flow**   Using the equation Eq. (9), the PDE that defines the Wasserstein gradient flow is

$$\frac{\partial q_t}{\partial t}(x) = -\nabla_x \cdot \left(q_t(x)\nabla_x \frac{p(x)}{q_t(x)}\right). \tag{90}$$

**Projected Wasserstein Gradient Flow**   Having the functional minimizing PDEs, we can find the update corresponding to the projected Wasserstein gradient flow using Proposition 3.2 and Eq. (17).

$$\Delta\theta^* \propto -\nabla_\theta \int q_t \left\langle \nabla_x \frac{\delta\mathcal{F}}{\delta q_t}, \nabla_x \log q(x,\theta)\right\rangle \mathrm{d}x \tag{91}$$

$$= \nabla_\theta \int q_t \left\langle \nabla_x \frac{p}{q_t}, \nabla_x \log q(x,\theta)\right\rangle \mathrm{d}x \tag{92}$$

$$= \nabla_\theta \int q_t \left\langle \frac{1}{q_t}\nabla_x p, \nabla_x \log q(x,\theta)\right\rangle \mathrm{d}x + \nabla_\theta \int q_t \left\langle p\frac{-1}{q_t^2}\nabla_x q_t, \nabla_x \log q(x,\theta)\right\rangle \mathrm{d}x \tag{93}$$

$$= \nabla_\theta \int p \left\langle \nabla_x \log p, \nabla_x \log q(x,\theta)\right\rangle \mathrm{d}x - \nabla_\theta \int p \left\langle \nabla_x \log q_t, \nabla_x \log q(x,\theta)\right\rangle \mathrm{d}x \tag{94}$$

$$= -\frac{1}{2}\nabla_\theta \mathbb{E}_{p(x)} \left[\left\|\nabla_x \log p(x) - \nabla_x \log q(x,\theta)\right\|^2\right]. \tag{95}$$

Note that similar to Proposition 3.2, we use $q_t(x) = q(x,\theta)$ as the density equal to the model density but detached from the parameters $\theta$. This objective corresponds to the score-matching objective (Hyvärinen & Dayan, 2005), which is widely used in the diffusion-based generative models (Song et al., 2020).

## C.2   Variational inference: forward KL divergence functional

Consider the forward KL divergence objective, and its first variation

$$\mathcal{F}[q_t] = D_{\mathrm{KL}}(q_t \,\|\, p) \tag{96}$$

$$\implies \frac{\delta\mathcal{F}}{\delta q_t} = 1 + \log\frac{q_t}{p}. \tag{97}$$

where $p(x)$ is the target distribution for variational inference.

**Fisher–Rao Gradient Flow**   Using equations Eq. (10), the PDE that defines the Fisher–Rao gradient flow is

$$\frac{\partial}{\partial t}q_t = -\mathrm{grad}_{\mathrm{FR}}\mathcal{F} \tag{98}$$

$$= q_t \left(-\frac{\delta\mathcal{F}}{\delta q_t} - \mathbb{E}_{q_t}\left[-\frac{\delta\mathcal{F}}{\delta q_t}\right]\right) \tag{99}$$

$$= q_t \left(-1 - \log\frac{q_t}{p} + \mathbb{E}_{q_t}\left[1 + \log\frac{q_t}{p}\right]\right) \tag{100}$$

$$= q_t \left(-\log\frac{q_t}{p} + D_{\mathrm{KL}}(q_t \,\|\, p)\right). \tag{101}$$

Thus we have

$$\frac{\partial}{\partial t} \log q_t = \log p - \log q_t + D_{\mathrm{KL}}(q_t \| p) \tag{102}$$

$$\implies \log q_t = \exp(-t) \log q_0 + (1 - \exp(-t)) \log p - \log \mathcal{Z}_t \tag{103}$$

$$\implies \boxed{q_t = \frac{1}{\mathcal{Z}_t} q_0^{\exp(-t)} p^{(1-\exp(-t))}} \qquad\qquad \mathcal{Z}_t = \int \mathrm{d}x \; q_0^{\exp(-t)} p^{(1-\exp(-t))}, \tag{104}$$

where $q_0$ is the initial distribution, and $\mathcal{Z}_t$ is the partition function. This results in a distributional path constructed by the geometric average of the end-point distributions. Note that changing the geometric parameter corresponds to *teleporting* particles. These distributional paths are commonly used in Annealed Importance Sampling (Neal, 2001).

**Projected Fisher–Rao Gradient Flow**  Having the functional minimizing PDEs, we can find the update corresponding to the projected Fisher–Rao gradient flow using Proposition 3.2 and Eq. (17),

$$\Delta\theta^* \propto - \int q_t \left(\frac{\delta\mathcal{F}}{\delta q_t}\right) \nabla_\theta \log q(x,\theta) \, \mathrm{d}x \tag{105}$$

$$= - \int q_t \left(1 + \log \frac{q_t}{p}\right) \nabla_\theta \log q(x,\theta) \, \mathrm{d}x \tag{106}$$

$$= - \int \log \frac{q_t}{p} \nabla_\theta q(x,\theta) \, \mathrm{d}x \tag{107}$$

$$= - \int \log q_t \nabla_\theta q(x,\theta) \, \mathrm{d}x + \nabla_\theta \int q(x,\theta) \log p \, \mathrm{d}x \tag{108}$$

$$= -\nabla_\theta \int q(x,\theta) \log q(x,\theta) \, \mathrm{d}x + \nabla_\theta \int q(x,\theta) \log p \, \mathrm{d}x \tag{109}$$

$$= -\nabla_\theta \int q(x,\theta) \log \frac{q(x,\theta)}{p} \, \mathrm{d}x \tag{110}$$

$$= -\nabla_\theta D_{\mathrm{KL}}\Big(q(x,\theta) \,\Big\|\, p\Big). \tag{111}$$

This corresponds to the standard variational inference objective.

**Wasserstein Gradient Flow**  Using the equation Eq. (9), the PDE that defines the Wasserstein gradient flow is

$$\frac{\partial}{\partial t} q_t = -\mathrm{grad}_{\mathrm{W}}\mathcal{F} \tag{112}$$

$$= \nabla_x \cdot \left(q_t \nabla_x \left(\frac{\delta\mathcal{F}}{\delta q_t}\right)\right) \tag{113}$$

$$= \nabla_x \cdot \left(q_t \nabla_x \left(1 + \log \frac{q_t}{p}\right)\right) \tag{114}$$

$$= \nabla_x \cdot (q_t \nabla_x (\log q_t - \log p)) \tag{115}$$

$$= -\nabla_x \cdot (q_t \nabla_x \log p) + \nabla \cdot (q_t \nabla_x \log q_t) \tag{116}$$

$$= -\nabla_x \cdot (q_t \nabla_x \log p) + \nabla_x^2 q_t \tag{117}$$

This is the Fokker–Planck equation, which characterize the movement of the particles according to Langevin dynamics:

$$\mathrm{d}X_t = \nabla_x \log p \, \mathrm{d}t + \sqrt{2}\mathrm{d}W_t \,.$$

This is a common approach for sampling from the unnormalized density $p(x)$ via Markov Chain Monte Carlo algorithms.

**Projected Wasserstein Gradient Flow**   Having the functional minimizing PDEs, we can find the update corresponding to the projected Wasserstein gradient flow using Proposition 3.2 and Eq. (17),

$$\implies \Delta\theta^* \propto -\nabla_\theta \int q_t \left\langle \nabla_x \frac{\delta\mathcal{F}}{\delta q_t}, \nabla_x \log q(x,\theta) \right\rangle dx \tag{118}$$

$$= -\nabla_\theta \int q_t \left\langle \nabla_x \left(1 + \log \frac{q_t}{p}\right), \nabla_x \log q(x,\theta) \right\rangle dx \tag{119}$$

$$= -\nabla_\theta \int q_t \left\langle \nabla_x \left(\log \frac{q_t}{p}\right), \nabla_x \log q(x,\theta) \right\rangle dx \tag{120}$$

$$= -\frac{1}{2} \mathbb{E}_{q(x,\theta)} \left[ \nabla_\theta \left\| \nabla_x \log q(x,\theta) - \nabla_x \log p \right\|^2 \right], \tag{121}$$

which resembles the score-matching objective (Hyvärinen & Dayan, 2005) for variational inference.

## D   c-Wasserstein gradient flow

$c$-Wasserstein distance with the convex cost function $c : \mathbb{R}^d \to \mathbb{R}$ is defined

$$W_c(p_0, p_1) = \inf_{\pi \in \Gamma(p_0, p_1)} \int \pi(x,y) c(x-y) \, dx \, dy, \tag{122}$$

where $\Gamma(p_0, p_1)$ is the set of all possible couplings of $p_0$ and $p_1$. The dynamic formulation of this distance is the following

$$W_c(p_0, p_1) := \inf_{v_t, q_t} \int_0^1 \mathbb{E}_{q_t(x)}[c(v_t(x))] \, dt, \quad \text{subj. to} \tag{123}$$

$$\frac{\partial q_t(x)}{\partial t} = -\nabla_x \cdot (q_t(x) v_t(x)), \quad \text{and} \quad q_0 = p_0, \; q_1 = p_1. \tag{124}$$

**Proposition.** *The energy-minimizing c-Wasserstein gradient flow is defined by the following PDE*

$$\frac{\partial q_t(x)}{\partial t} = -\nabla_x \cdot (q_t(x) \nabla c^*(-\nabla_x E_{\mathrm{loc}}(x))), \tag{125}$$

*where $c^*(\cdot)$ is the convex conjugate function of $c(\cdot)$.*

*Proof.* The movement minimizing scheme for $W_c(\cdot, \cdot)$ is the following optimization problem

$$\inf_{q'} F[q'] - F[q] + \frac{1}{\Delta t} W_c(q, q'). \tag{126}$$

Assuming that the density changes according to the continuity equation $q'(x) = q_t(x) - \Delta t \nabla_x \cdot (q_t(x) v_t(x)) + o(\Delta t)$, and $\Delta t$ is small enough so that $v_t(x)$ defines the optimal transportation plan, we have

$$\inf_{v_t} F[q_t] - \Delta t \int \nabla_x \cdot (q_t(x) v_t(x)) \frac{\delta F[q_t]}{\delta q_t}(x) \, dx - F[q_t] + \frac{1}{\Delta t} \Delta t^2 \mathbb{E}_{q_t(x)} c(v_t(x)) \tag{127}$$

$$= \Delta t \inf_{v_t} \int q_t(x) \left\langle v_t(x), \nabla_x \frac{\delta F[q_t]}{\delta q_t}(x) \right\rangle dx + \mathbb{E}_{q_t(x)} c(v_t(x)) \tag{128}$$

$$= \Delta t \inf_{v_t} \int q_t(x) \left[ c(v_t(x)) - \left\langle v_t(x), -\nabla_x \frac{\delta F[q_t]}{\delta q_t}(x) \right\rangle \right] dx \tag{129}$$

$$= -\Delta t \int q_t(x) c^* \left( -\nabla_x \frac{\delta F[q_t]}{\delta q_t}(x) \right) dx \tag{130}$$

and the infimum is achieved at

$$v_t(x) = \nabla c^* \left( -\nabla_x \frac{\delta F[q_t]}{\delta q_t}(x) \right), \tag{131}$$

which gives the formula for the vector field. Using the energy gradient from Theorem 3.1 $\frac{\delta E[q_t]}{\delta q_t}(x) = E_{\mathrm{loc}}$, we get the result.

$\square$

**Proposition.** *Coordinate-wise application of* $\tanh$ *to the vector field, i.e.*

$$\frac{\partial q_t(x)}{\partial t} = -\nabla_x \cdot (q_t(x) \tanh(-\nabla_x E_{\text{loc}}(x)))\,, \tag{132}$$

*corresponds to gradient descent with c-Wasserstein distance, where $c : \mathbb{R}^d \to \mathbb{R}$ is the following cost function*

$$c(x) = \sum_i^d \frac{1}{2}\Big((x_i + 1)\log(x_i + 1) + (1 - x_i)\log(1 - x_i)\Big) - d\log 2. \tag{133}$$

*Proof.* Consider $c^*(x) = \sum_i \log(\exp(x_i) + \exp(-x_i))$. It corresponds to applying hyperbolic tangent non-linearity coordinate-wise to the vector field field, i.e.,

$$\partial_i c^*(x) = \frac{\exp(x_i) - \exp(-x_i)}{\exp(x_i) + \exp(-x_i)} = \tanh(x_i). \tag{134}$$

The corresponding cost function $c(x)$ is the following

$$c(x) = \sup_y \langle x, y \rangle - c^*(y) \tag{135}$$

$$= \sup_y \sum_i^d \Big(x_i y_i - \log(\exp(y_i) + \exp(-y_i))\Big) \tag{136}$$

$$= \sup_y \sum_i^d \Big((x_i + 1)y_i - \log(\exp(2y_i) + 1)\Big) \qquad //y_i = \frac{1}{2}\log\frac{1 + x}{1 - x} \tag{137}$$

$$= \sum_i^d \left((x_i + 1)\frac{1}{2}\log\left(\frac{1 + x_i}{1 - x_i}\right) - \log\left(\frac{1 + x_i}{1 - x_i} + 1\right)\right) \tag{138}$$

$$= \sum_i^d \frac{1}{2}\Big((x_i + 1)\log(x_i + 1) + (1 - x_i)\log(1 - x_i)\Big) - d\log 2. \tag{139}$$

$\square$

# E  Additional experimental results

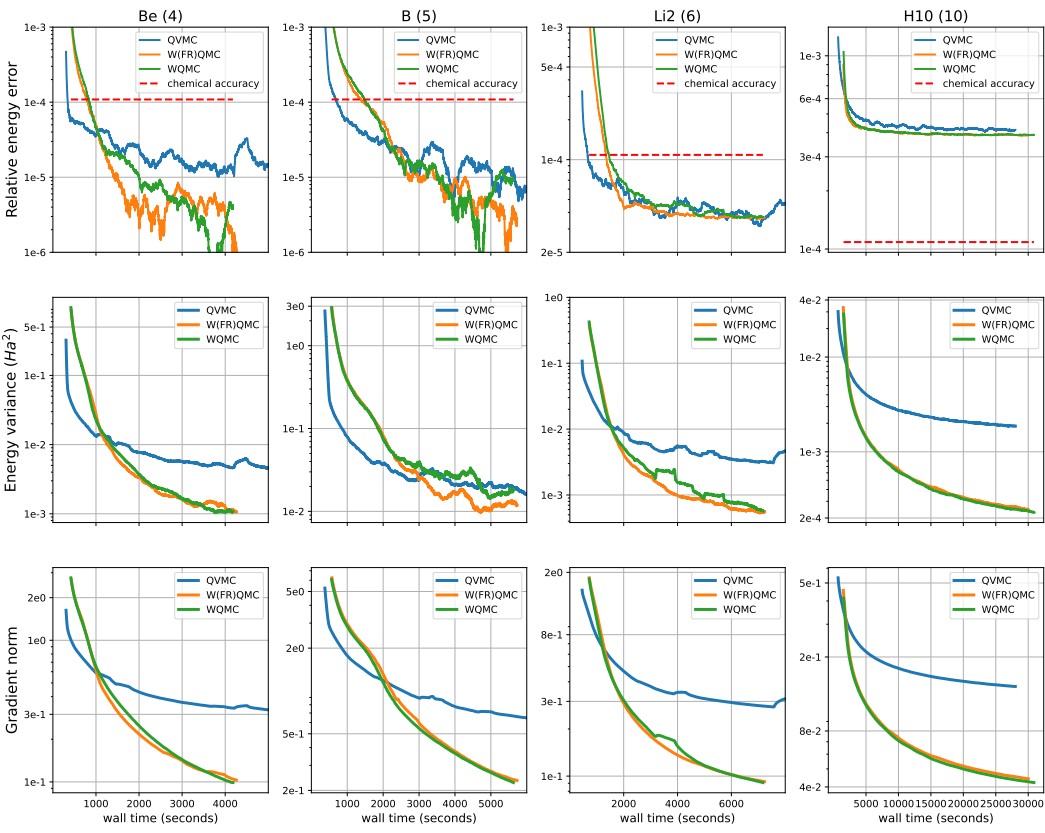

Figure 3: Optimization results for different chemical systems (every column corresponds to a given molecule). The number of electrons is given in the brackets next to systems' names. Throughout the optimization, we monitor three values: the mean value of the local energy (lower is better), the variance of the local energy, and the median value of the gradient norm of the local energy. In the first row of plots, we average (removing $5\%$ of outliers from both sides) the energy over $1000$ iterations and report the relative error to the actual ground-state energy: $(E - E_0)/E_0$. In the second row, we report standard deviation averaged over $1000$ iterations (removing $5\%$ of outliers from both sides). In the third row, we report the median gradient norm averaged over $1000$ iterations (removing $5\%$ of outliers from both sides). All the metrics are plotted versus the wall time of computations performed on four Nvidia A40 GPUs.