# OpenReview forum: "Wasserstein Quantum Monte Carlo: A Novel Approach for Solving the Quantum Many-Body Schrödinger Equation"
_NeurIPS.cc/2023/Conference — NeurIPS 2023 spotlight_

### Official Review · Reviewer_Qbaw · 2023-06-23

**Soundness:** 2 fair
**Presentation:** 3 good
**Contribution:** 3 good
**Rating:** 6
**Confidence:** 3

**Summary:**

The paper provides a framework to derive a new loss for Variational Monte Carlo methods. They introduce Wasserstein Quantum Monte Carlo, which uses a gradient flow based on the Wasserstein metric. The method is empirically tested for fermionic systems, e.g. a Hydrogen chain and compared against another Quantum Monte Carlo approach based on the Fisher-Rao metric. Faster convergence in terms of number of epochs is observed.

**Strengths:**

The paper derives with a theoretical framework a new loss function for Quantum Monte Carlo calculations.
They underline their findings with experiments and improve upon state-of-the-art variational Monte Carlo calculations with respect to the number of epochs on rather small molecular systems.
A lot of recent improvements in this field came mainly from architectural changes (e.g. PsiFormer), it is great to see also improvements on the loss function and optimization side.


**Weaknesses:**

The derived loss function is novel and improves with respect to the number of epochs current state-of-the-art calculation, I have the following concerns:

1.	The lack of immediate relevance to the ML community, due to the rather small experimental section and the newly derived loss fits rather to the Quantum Chemistry community.
2.	What is the QVMC baseline in section 4, is it PsiFormer? What kind of settings did you use for the PsiFormer architecture? The authors of PsiFormer proposed two variants, a small and a larger one.
3.	The loss WQMC loss seems to converge faster in terms of epochs but based on the Algorithm 1 my understanding is that you have to perform third derivatives which is computationally quite expensive. Could the authors add to Fig. 2 the timings per epoch for each method and the type of GPUs used to perform the experiments?
4.	In case it is possible it would be also interesting to see a broader experimental section rather than these “simple” molecules to better understand the improvements gained through the newly defined loss. For example, the potential energy surface of the Nitrogen dimer is a difficult system (especially considering the bond breaking geometry), which was also analyzed by the authors from FermiNet.


**Questions:**

1.	In algorithm 1 you stop the gradient for the gradient of the local energies. It is not clear to me from eq. 24, why this is the case?

**Limitations:**

All limitations are addressed.

---

> ### Author Rebuttal · Authors · 2023-08-10
>
> Thank you for your valuable feedback and the time spent. We are glad that you appreciate our developments in the design of new optimization procedures for variational Monte Carlo methods. In what follows we address the concerns raised and answer the questions.
>
> > The lack of immediate relevance to the ML community, due to the rather small experimental section and the newly derived loss fits rather to the Quantum Chemistry community.
>
> We respectfully disagree. The NeurIPS community has a rich tradition of ML applications to other fields of science including natural sciences, e.g. numerous AI4Science workshops evidence for this (e.g., see NeurIPS 2022 "Machine Learning and the Physical Sciences" where many papers including the opening keynote were about quantum variational Monte Carlo). Moreover, different approaches to this particular problem were recently published at the top ML conferences including NeurIPS [1,2,3]. While in this paper, we introduced our approach in the context of energy functional minimization for quantum systems, it is quite general-purposed: it can be applied to minimize any functional directly in the space of distributions, making it relevant to the wider audience within the NeurIPS community. For example, instead of using the quantum energy functional one might consider the KL divergence functional $\text{KL}(p \\| q)$ between the data distribution $p$ and the model distribution $q$. Then the Fisher--Rao metric corresponds to the energy-based model training and the Wasserstein metric corresponds to score-matching. We will draw these connections and expand on this discussion in the next revision of the paper.
>
> > What is the QVMC baseline in section 4, is it PsiFormer? What kind of settings did you use for the PsiFormer architecture? The authors of PsiFormer proposed two variants, a small and a larger one.
>
> For the baseline, we precisely follow [3]. Thus, as considered in [3] for the systems of similar size, we use small Psiformer architecture, the same hyperparameters, and the same techniques to stabilize optimization.
>
> > The loss WQMC loss seems to converge faster in terms of epochs but based on the Algorithm 1 my understanding is that you have to perform third derivatives which is computationally quite expensive. Could the authors add to Fig. 2 the timings per epoch for each method and the type of GPUs used to perform the experiments?
>
> As suggested by the reviewer, for a proper comparison, we updated Fig. 2 with wall-clock time as the x-axis and uploaded the new figure in our global response. The type of GPUs used to run experiments is also mentioned in the global response. In short, all the claims of our paper still hold when considering wall-clock time. Third-order derivatives can be efficiently computed using modern deep learning frameworks such as JAX, which we used to implement our method. We will include these updates in the final paper.
>
> > In case it is possible it would be also interesting to see a broader experimental section rather than these “simple” molecules to better understand the improvements gained through the newly defined loss. For example, the potential energy surface of the Nitrogen dimer is a difficult system (especially considering the bond breaking geometry), which was also analyzed by the authors from FermiNet.
>
> The systems considered in the experiments are not “simple”. Indeed, a single run for the H10 system takes more than 33 GPU hours on Nvidia A40 GPU, and we can see that there is already a gap between the baseline accuracy and the chemical accuracy. In this work, we focus on extending the Variational Monte Carlo algorithms family, and illustrating that the proposed extensions might be indeed beneficial for the performance. Scaling the proposed method to bigger molecules requires a more comprehensive empirical study and substantial engineering infrastructure, which are beyond the scope of this project.
>
> > In algorithm 1 you stop the gradient for the gradient of the local energies. It is not clear to me from eq. 24, why this is the case?
>
> Thank you for bringing this up! The local energy in (24) is evaluated at $q_t(x) = q(x,\theta)$ but does not depend on parameters $\theta$. Indeed equation (24) is obtained by using (23) for the density time derivative in (17) and integration by parts. Note that the gradient w.r.t. $\theta$ is applied only to the log density in (17) and to the gradient of log density after the integration by parts. We will further clarify this in the next revision of the paper.
>
> [1] Gerard, Leon, et al. "Gold-standard solutions to the Schrödinger equation using deep learning: How much physics do we need?." Advances in Neural Information Processing Systems 35 (2022): 10282-10294.
> [2] Gao, Nicholas, and Stephan Günnemann. "Generalizing neural wave functions.", International Conference on Machine Learning 2023.
> [3] von Glehn, Ingrid, James S. Spencer, and David Pfau. "A Self-Attention Ansatz for Ab-initio Quantum Chemistry." The Eleventh International Conference on Learning Representations. 2022.

---

> > ### Comment · Reviewer_Qbaw · 2023-08-11
> >
> > Thank you for your detailed response! Based on the rebuttal, I will revise my score and recommend a weak accept.
> >
> > > Immediate relevance to ML community
> >
> > Thank you for clarifying this and I will recommend the paper to be accepted.
> >
> > > Experimental section
> >
> > I still disagree that the tested systems are a good set of systems to showcase their new approach but I understand that a detailed experimental section is out of reach for this work and maybe not the focus of the work. Although, I believe it would improve the paper significantly, especially the magical threshold from chemistry of chemical accuracy is faster reached for the rather small systems with the standard approach (wrt. wall clock time). Here, more complex systems would potentially showcase the importance of having a better optimisation scheme.
> >
> > I admit the results are impressive, especially gradient norm and variance, and I appreciate the additional figure regarding wall clock time.
> >
> > > Another issue, common to all Variational Monte Carlo methods, is the singularities around nuclei, which create instabilities during training.
> >
> > Could you elaborate on this comment a little bit more, my understanding for the tested systems (Be, B, Li2, H10) is that the current architectures like FermiNet and PsiFormer are quite stable.
> > Do you observe divergences of the energy during training? You mention effective core potential as a potential solution, do you expect effective core potential to be already necessary for atoms like Oxygen, Nitrogen or Carbon or just for heavy atoms (4th row and so on)?
> >
> > ### Minor questions
> >
> > 1. Are the blue curves the same in the main text and the additional pdf (belonging to the rebuttal)? If I compare for B in the opt. step plot and in the wall clock time plot the blue curve they seem to be different, whereas for the orange and the green curve they seem to be identical. Or is this a plotting artifact?
> >
> > 2. Do you assume your new optimization scheme to scale worse than the standard approach with the number of electrons?

---

> > > ### Author Response · Authors · 2023-08-11
> > >
> > > Thank you for carefully going over the rebuttal and reconsidering your evaluation! We appreciate your time and efforts spent.
> > >
> > > > Do you observe divergences of the energy during training? You mention effective core potential as a potential solution, do you expect effective core potential to be already necessary for atoms like Oxygen, Nitrogen or Carbon or just for heavy atoms (4th row and so on)?
> > >
> > > In our comment, we refer to cusps instabilities, which are irrelevant to the proposed methodology and are common to both the baseline and the proposed method. Note that this is related mostly to the architectural design choice. Indeed, Psiformer [1] architecture does not ensure any cusp conditions by design and the authors discuss different ways to stabilize the training (see Fig. 3,5, appendices 1, 2.c, 2.d in [1]). Alternatively, one can ensure cusp conditions by design, e.g. see PauliNet architecture [2]. We are sorry for the possible confusion.
> > >
> > > The introduction of pseudopotentials is a common technique for dealing with the cusps/singularities. We have not experimented with it, but one can find open-source solutions in the DeepQMC framework [3]. Also, we refer the reader to [3] for the empirical study and the guidelines for using the pseudopotential.
> > >
> > > > Are the blue curves the same in the main text and the additional pdf (belonging to the rebuttal)?
> > >
> > > As we discussed in our previous response, the proposed method takes more time per iteration. So in order to have a proper runtime comparison, we re-ran the entire experiment for QVMC with more total iterations, and thus the entire blue curve is new. However, the general pattern of the new blue curve closely matches the old blue curve after rescaling the x-axis. The remaining curves are identical.
> > >
> > > > Do you assume your new optimization scheme to scale worse than the standard approach with the number of electrons?
> > >
> > > In our experiments, we observe the biggest performance gain for the Hydrogen chain system (H10). Hence, we do not expect our method to scale worse than the baseline with the number of electrons. Moreover, as we discuss in the conclusion (lines 290-297), we expect our method to scale better for multi-modal target distributions (e.g. due to multiple atoms in molecular systems) due to constraining the density model updates to local changes, which favor faster MCMC mixing.
> > >
> > > [1] von Glehn, Ingrid, James S. Spencer, and David Pfau. "A Self-Attention Ansatz for Ab-initio Quantum Chemistry." The Eleventh International Conference on Learning Representations. 2022.
> > > [2] Hermann, Jan, Zeno Schätzle, and Frank Noé. "Deep-neural-network solution of the electronic Schrödinger equation." Nature Chemistry 12.10 (2020): 891-897.
> > > [3] Schätzle, Zeno, et al. "DeepQMC: an open-source software suite for variational optimization of deep-learning molecular wave functions." arXiv preprint arXiv:2307.14123 (2023).

---

### Official Review · Reviewer_Nk4c · 2023-06-28

**Soundness:** 4 excellent
**Presentation:** 4 excellent
**Contribution:** 3 good
**Rating:** 7
**Confidence:** 3

**Summary:**

The authors show that the optimization objective typically used in variational Monte Carlo (VMC) solutions of the Schrödinger equation can be interpreted as the Fisher-Rao gradient flow in the space of Born distributions. Based on this insight, they suggest to substitute the Fisher-Rao metric with Wasserstein metrics, which has the effect that during optimization, probability mass is "transported gradually" rather than "teleported" in space. Since the VMC optimization process typically relies on sampling "walkers" from the probability distribution, which evolve over time according to a Markov process, a gradual "transport" of the probability mass promises improved convergence. The authors empirically demonstrate that their proposed method leads to lower ground state energies (and variance) for several small systems.

**Strengths:**

The paper introduces a very simple (although non-trivial!) idea and is clearly presented. The proposed optimization methods show clear practical advantages and the underlying theory may lead to even more sophisticated optimization methods for VMC solutions of the Schrödinger equation in the future.

**Weaknesses:**

The empirical evaluation of the proposed method is limited to relatively few and small systems. While the results are very promising, it is therefore unclear how the proposed method compares to the standard approach when applied to more complicated, larger systems (e.g. benzene). This being said, I find this limitation acceptable considering the large computational cost of running VMC calculations for larger systems (which the authors may not be able to afford).

**Questions:**

- Have the authors considered applying their method also to larger test systems (e.g. benzene), or even just small molecular test systems (say ethanol, methane, ...)? Currently the method is only tested for two atoms, the lithium dimer, and the H10 chain. I encourage the authors to extend their empirical evaluation to more complicated examples (if they can afford it).

- The plots in Figure 2 show the number of iterations, but they do not show the associated runtime costs. Is a single iteration of WQMC comparable in cost to a single iteration of QVMC?

- A single iteration of VMC consists of several individual steps including the parameter update followed by sampling walker positions for the updated probability density. The sampling of walker positions typically uses a certain number of steps in a Markov process. Considering the "more well-behaved" updates of the probability density with WQMC compared to QVMC, is it possible to reduce the required number of sampling steps for the walker positions? If the authors have not considered this, I encourage them to try. Perhaps it is possible to decrease the runtime costs of a single iteration this way?

**Limitations:**

The authors do not discuss limitations of their work, nor its potential for negative societal impact. While the latter is acceptable in my opinion (this work is about an algorithmic method to solve a fundamental problem in quantum chemistry and a discussion of negative societal impacts would probably appear contrived), a discussion of limitations should be added in a revised version.

---

> ### Author Rebuttal · Authors · 2023-08-10
>
> Thank you for your insightful and positive feedback. We are glad that you appreciated both the simplicity and non-trivial nature of our idea. We also believe that our underlying theory has many potential implications for developing even better VMC solutions. In what follows we address the raised concerns and answer your questions.
>
> > The plots in Figure 2 show the number of iterations, but they do not show the associated runtime costs. Is a single iteration of WQMC comparable in cost to a single iteration of QVMC?
>
> We include the runtime of all the algorithms in the plots (see attachment in the general response). Namely, for a proper comparison, instead of reporting the metrics per iteration we report them per seconds elapsed. We will include these plots in the next version of the paper. All the models were benchmarked on four A40 GPUs.
>
> > Considering the "more well-behaved" updates of the probability density with WQMC compared to QVMC, is it possible to reduce the required number of sampling steps for the walker positions? If the authors have not considered this, I encourage them to try. Perhaps it is possible to decrease the runtime costs of a single iteration this way?
>
> In our experiments, we observed that the proposed method indeed can be trained with fewer MCMC steps (not included in the paper). Moreover, one can use the evaluated vector field (gradient of the local energy) as the first proposal update to the samples. We leave the detailed study of this degree of freedom beyond the scope of the current work, instead, we apply minimal changes to the optimization procedure to highlight the effect of the newly introduced metric.
>
> > I encourage the authors to extend their empirical evaluation to more complicated examples (if they can afford it).
>
> The systems considered in the experiments are not “simple”. Indeed, a single run for the H10 system takes more than 33 GPU hours on Nvidia A40 GPU, and we can see that there is already a gap between the baseline accuracy and the chemical accuracy. In this work, we focus on extending the Variational Monte Carlo algorithms family, and illustrating that the proposed extensions might be indeed beneficial for the performance. Scaling the proposed method to bigger molecules requires a more comprehensive empirical study and substantial engineering infrastructure, which are beyond the scope of this project.
>
> > Discussion of limitations should be added in a revised version
>
> As suggested by the reviewer, we will include a limitation section in the final paper. The main downside of the proposed method is the additional evaluation of the derivative, which results in longer iterations. Another issue, common to all Variational Monte Carlo methods, is the singularities around nuclei, which create instabilities during training. In the general response, we discuss how these limitations can be addressed in future work.

---

> > ### Comment · Reviewer_Nk4c · 2023-08-13
> >
> > I thank the authors for their detailed reply. Regarding my comment regarding the "simplicity" of the considered systems: I understand that there are different opinions on what constitutes a "simple" system. With "simple" I mainly meant the number of electrons. For example, H10 with just 10 electrons is simply not a particularly large example, but I am aware that it is still challenging for many methods due to its highly correlated character. I also do not consider 33 GPU hours to be a particularly large investment of computational resources, but I understand that access to computational resources is very inhomogeneous in the field. I still find a larger test system would make the paper more interesting, but again, I leave it up to the authors whether they want to consider this.

---

### Official Review · Reviewer_n7Uw · 2023-07-06

**Soundness:** 3 good
**Presentation:** 3 good
**Contribution:** 3 good
**Rating:** 5
**Confidence:** 1

**Summary:**

The authors propose a novel approach named Wasserstein Quantum Monte Carlo, which uses the gradient flow induced by the Wasserstein and Wasserstein Fisher-Rao metrics and improves the convergence rate of Quantum Variational Monte Carlo. The numerical results show that following the gradient flow under Wasserstein and Wasserstein Fisher-Rao metrics results in better convergence to the ground state wave function.

**Strengths:**

1. This paper is technically sound with detailed derivations.
2. Seeking new approaches to better solve quantum chemistry and quantum physics problems is very important to the quantum machine learning society.

**Weaknesses:**

I'm not familiar with this subject but I strongly believe that this paper belongs to a Physics conference or journal instead of NeurIPS.

**Questions:**

The numerical results show that the proposed method outperforms the QVMC. But it seems not that good compared to variational quantum eigensolvers (such as UCCSD or adaptVQE) on the ground-state energy estimation problems. Could you illustrate the difference between the proposed method and those VQE methods?

**Limitations:**

Wrong track to submit this paper to NeurIPS.

---

> ### Author Rebuttal · Authors · 2023-08-10
>
> Thank you for your time spent and valuable feedback.
>
> > I'm not familiar with this subject but I strongly believe that this paper belongs to a Physics conference or journal instead of NeurIPS
>
> We respectfully disagree that this paper does not belong to NeurIPS. The NeurIPS community has a rich tradition of ML applications to other fields of science including natural sciences, e.g. numerous AI4Science workshops evidence for this (e.g., see NeurIPS 2022 "Machine Learning and the Physical Sciences" where many papers including the opening keynote were about quantum variational Monte Carlo). Moreover, different approaches to this particular problem were recently published at the top ML conferences including NeurIPS [1,2,3]. While in this paper, we introduced our approach in the context of energy functional minimization for quantum systems, it is quite general-purposed: it can be applied to minimize any functional directly in the space of distributions, making it relevant to the wider audience within the NeurIPS community. For example, instead of using the quantum energy functional one might consider the KL divergence functional $\text{KL}(p \Vert q)$ between the data distribution $p$ and the model distribution $q$. Then the Fisher--Rao metric corresponds to the energy-based model training and the Wasserstein metric corresponds to score-matching. We will draw these connections and expand on this discussion in the next revision of the paper.
>
> > Could you illustrate the difference between the proposed method and those VQE methods?
>
> The considered Variational Monte Carlo methods (both the baseline and the proposed method) are classical computation approaches to the quantum chemistry problem. On the other side, (adapt)VQE is an algorithm for quantum devices, and cannot be efficiently run until efficient quantum devices are available. Hence, we provide no comparison. If compared to CCSD, which is the classical version of UCCSD, the approaches considered in the paper are known to be much more scalable (see Fig. 1 in [4]).
>
> [1] Gerard, Leon, et al. "Gold-standard solutions to the Schrödinger equation using deep learning: How much physics do we need?." Advances in Neural Information Processing Systems 35 (2022): 10282-10294.
> [2] Gao, Nicholas, and Stephan Günnemann. "Generalizing neural wave functions.", International Conference on Machine Learning 2023.
> [3] von Glehn, Ingrid, James S. Spencer, and David Pfau. "A Self-Attention Ansatz for Ab-initio Quantum Chemistry." The Eleventh International Conference on Learning Representations. 2022.
> [4] Hermann, Jan, et al. "Ab-initio quantum chemistry with neural-network wavefunctions." arXiv preprint arXiv:2208.12590 (2022).

---

> > ### Comment · Reviewer_n7Uw · 2023-08-18
> >
> > Thanks for clarifying. Again, I'm not very familiar with this topic and I have given a very low confidence about my evaluation. I hope this will not affect the overall rating of this paper.

---

### Official Review · Reviewer_f58M · 2023-07-16

**Soundness:** 3 good
**Presentation:** 3 good
**Contribution:** 3 good
**Rating:** 7
**Confidence:** 3

**Summary:**

The paper proposes a new way to compute gradients for parameters in quantum Monte Carlo. The authors propose to update the network parameters by following the Wasserstein Fishier-Rao gradient flow, which is composed of a continuity equation and a  growth term. The authors show that when only the latter term is considered, the WFR gradient flow is equivalent the conventional energy loss. The authors then propose to leverage the first term and propose the Wasserstein quantum Monte Carlo. With some tricks the proposed optimization method leads to a faster convergence. Experiments are conducted on small systems with atoms.

**Strengths:**

- The paper is very well written.
- The proposed method is shown to include the conventional method.
- The experimental results are improved.

**Weaknesses:**

- It's not explicitly mentioned in the paper. But I think the extra gradient computation may cause more computation.

**Questions:**

- Since some tricks are used in experiments. How does these tricks influence the performance? Are those tricks also used for baseline methods?
-  Which would be the recommended method? The WQMC or the W(FR)QMC? What would be the trade-off?

---

> ### Author Rebuttal · Authors · 2023-08-10
>
> Thank you for your valuable feedback and time spent. We are glad you found our paper well-written. In what follows we address the concerns raised and answer the questions.
>
> > It's not explicitly mentioned in the paper. But I think the extra gradient computation may cause more computation.
>
> Indeed, the proposed method uses extra gradient computation. We will clarify it in the paper. In the general response, we added the plots comparing the methods in terms of runtime instead of iterations to take this extra computation into account. Note that all the claims in the paper still hold in terms of wall time. We will include these plots in the next version of the paper.
>
> > Since some tricks are used in experiments. How does these tricks influence the performance? Are those tricks also used for baseline methods?
>
> Filtering out outliers and clipping the gradients are standard practices in variational Monte Carlo methods for molecules. For the baseline, we precisely follow [1] where different clippings were used to stabilize the optimization procedure. For the proposed method, we design analogs for the new objective and use the same tricks as proposed in [1] where it is possible.
>
> > Which would be the recommended method? The WQMC or the W(FR)QMC? What would be the trade-off?
>
> We observe that the Fisher-Rao metric demonstrates faster convergence in the beginning and the Wasserstein metric converges faster later into the optimization. Our intuition is that the usage of both metrics at the same time allows us adaptively decide which distance to the ground state is larger at the moment and minimize it.
> Indeed, one can introduce a hyperparameter controlling the tradeoff between metrics. In our experiments, we haven’t investigated the influence of this hyperparameter, and simply take the sum of two metrics for W(FR)QMC. We leave an extensive study of this parameter for future work.
>
> [1] von Glehn, Ingrid, James S. Spencer, and David Pfau. "A Self-Attention Ansatz for Ab-initio Quantum Chemistry." The Eleventh International Conference on Learning Representations. 2022.

---

> > ### Comment · Reviewer_f58M · 2023-08-15
> >
> > Thanks for the clarifications. I will raise my score.

---

### Official Review · Reviewer_wiqn · 2023-07-22

**Soundness:** 3 good
**Presentation:** 4 excellent
**Contribution:** 4 excellent
**Rating:** 8
**Confidence:** 4

**Summary:**

The paper shows that quantum variational Monte Carlo can be written as energy minimizing gradient flow under Fisher-Rao metric, and proposes an alternative to traditional quantum Monte Carlo by following Wasserstein gradient flow. Experiments with Be, B, Li2, and H10 chain show lower energies achieved by including Wasserstein gradient flow.

**Strengths:**

Paper proposes a novel optimization approach to quantum Monte Carlo, an important problem for computational physics/chemistry. Presentation and writing is very clear, mathematically rigorous. The results show better convergence and better optima reached with proposed method.

**Weaknesses:**

only 4 physical systems and one NN architecture were tested.
Paper seems to state that WQMC is better for all cases; if so, more experiments would make this claim stronger.


**Questions:**

Why does W(FR)QMC seem to be better than WQMC alone for most of the experiments? What was the hyper-parameter controlling the W/FR tradeoff, and how sensitive is this hyper-parameter?

Were other c- cost functions tried for the c-Wasserstein gradient flow? why does the coordinate-application of tanh work well?

stop_gradient in algorithm box was referring to ?

**Limitations:**

Do authors expect including WQMC to converge better for all physical systems? For ex, Does the strong/weak correlation of the physics affect the method convergence relative to QMC?

Could authors include the additional computation complexity (theoretically, empirically) required for the Wasserstein gradient ?

---

> ### Author Rebuttal · Authors · 2023-08-10
>
> Thank you for your valuable feedback and time spent. We are glad that you find the presentation of our ideas to be clear and rigorous. In what follows we answer the questions asked.
>
> > Why does W(FR)QMC seem to be better than WQMC alone for most of the experiments? What was the hyper-parameter controlling the W/FR tradeoff, and how sensitive is this hyper-parameter?
>
> We observe that the Fisher-Rao metric demonstrates faster convergence in the beginning and the Wasserstein metric converges faster later into the optimization. Our intuition is that the usage of both metrics at the same time allows us adaptively decide which distance to the ground state is larger at the moment and minimize it.
> Indeed, one can introduce a hyperparameter controlling the tradeoff between metrics. In our experiments, we haven’t investigated the influence of this hyperparameter, and simply take the sum of two metrics for W(FR)QMC. We leave an extensive study of this parameter for future work.
>
> > Were other c- cost functions tried for the c-Wasserstein gradient flow? why does the coordinate-application of tanh work well?
>
> In our experiments, we concluded that for stable training it’s important that the c-cost function clips high values of the vector field. Our intuition is that it is related to the instabilities created by cusps close to nuclei, which are common to all variational Monte Carlo methods. We also experimented with the sign non-linearity, i.e. applying it coordinate-wise to the local energy gradient. Although it exhibits faster convergence at the beginning of the optimization, we find that it demonstrates higher variance later on.
>
> > stop_gradient in algorithm box was referring to?
>
> For better presentation, and to provide some intuition on the implementation, by stop_gradient we denote the operation that detaches the node from the computational graph. That is, we evaluate the value of the local energy gradient, but then we only use its value, not backpropagating w.r.t. inputs or parameters so it’s not affected by other gradient operators. We will clarify this in the next revision of the paper, thank you!
>
> > Do authors expect including WQMC to converge better for all physical systems? For ex, Does the strong/weak correlation of the physics affect the method convergence relative to QMC?
>
> Although our approach is agnostic to the functional (and, correspondingly, Hamiltonian), we expect our method to improve over traditional QMC as the effect of correlations increases. In the limit where the system becomes non-interacting, a single Slater determinant can represent the ground state and hence consists of single-particle orbitals only. More importantly, however, the cases we expect our methodology to work best in, are where the target Born distribution is multi-modal, e.g. due to multiple atoms in molecular systems. Due to the latter, our significantly improved results for H10 are not unexpected.
>
> > Could authors include the additional computation complexity (theoretically, empirically) required for the Wasserstein gradient?
>
> As requested by most of the reviewers, we include the runtime of all the algorithms in the plots (see the new plots in the pdf attached to the general response). Namely, for a proper comparison, instead of reporting the metrics per iteration we report them per wall time in seconds. Note that all the claims in the paper still hold in terms of wall time. We will include these plots in the next version of the paper. All the models were benchmarked on four A40 GPUs.

---

> > ### Comment · Reviewer_wiqn · 2023-08-16
> >
> > Hi Authors, thanks for your response, and I am happy with the discussion.

---

### Author Rebuttal · Authors · 2023-08-10

We thank all reviewers for their time spent and valuable feedback on the paper. In what follows we would like to address the common concerns raised by most of the reviewers.

**Runtime.** As requested by most of the reviewers, we include the runtime of all the algorithms in the plots (see the new plots in the attached pdf to this response). Namely, for a proper comparison, instead of reporting the metrics per iteration we report them per wall time in seconds. Note that all the claims in the paper still hold in terms of wall time. We will include these plots in the next version of the paper. All the models were benchmarked on four A40 GPUs.

**Limitations.** As suggested by reviewer Nk4c, we will include a limitation section in the final paper. The main downside of the proposed method is the additional evaluation of the derivative, which results in longer iterations. Another issue, common to all Variational Monte Carlo methods, is the singularities around nuclei, which create instabilities during training.

**Future work.** We believe that the aforementioned limitations can be addressed in future work with an extensive empirical study.
1. One direction is to come up with more efficient Monte Carlo schemes (updates of the samples). Indeed, in our experiments, we observed that the proposed method indeed requires fewer MCMC steps (not included in the paper). Moreover, one can use the evaluated vector field (gradient of the local energy) as the first proposal update to the samples. This would require additional tuning of the hyperparameters (such as the proposal step) but would allow to decrease the runtime.
2. Another direction is the usage of the effective core potentials for larger atoms, which help alleviate instabilities created by cusps around nuclei (see, e.g. [1]).
3. Finally, one can study alternative metrics by choosing the tradeoff between Fisher-Rao and Wasserstein metrics. In our experiments, we observe that such a tradeoff exists since the joint metric W(FR)QMC (without tuned coefficient) stably outperforms both metrics used separately.

**Scaling experiments.** The systems considered in the experiments are not “simple”. Indeed, a single run for the H10 system takes more than 33 GPU hours on Nvidia A40 GPU, and we can see that there is already a gap between the baseline accuracy and the chemical accuracy. In this work, we focus on extending the Variational Monte Carlo algorithms family, and illustrating that the proposed extensions might be indeed beneficial for the performance. Scaling the proposed method to bigger molecules requires a more comprehensive empirical study and substantial engineering infrastructure, which are beyond the scope of this project.

[1] Li, Xiang, et al. "Fermionic neural network with effective core potential." Physical Review Research 4.1 (2022): 013021.

---

### Decision · Program_Chairs · 2023-09-21

**Decision:**

Accept (spotlight)

**Comment:**

This paper provides new insights into quantum variational Monte Carlo (QVMC) by interpreting QVMC as a different objective, leading to alternative QMC algorithms based on Wasserstein gradient flow. Overall, reviewers came to a consensus about accepting this paper and appreciated the clear presentation and the empirical results presented with the proposed approach. Finally, reviewers noted that in addition to the method being a interesting contribution on its own, the theoretical insights from this paper may also lead to the development of novel VMC algorithms.